JCB Journal of Cell Biology

# UBAP2L contributes to formation of P-bodies and modulates their association with stress granules

Claire L. Riggs[1,2], Nancy Kedersha[1,2], Misheel Amarsanaa[1,2,3], Safiyah Noor Zubair[1,2], Pavel Ivanov[1,2], and Paul Anderson[1,2]

**Stress triggers the formation of two distinct cytoplasmic biomolecular condensates: stress granules (SGs) and processing bodies (PBs), both of which may contribute to stress-responsive translation regulation. Though PBs can be present constitutively, stress can increase their number and size and lead to their interaction with stress-induced SGs. The mechanism of such interaction, however, is largely unknown. Formation of canonical SGs requires the RNA binding protein Ubiquitin-Associated Protein 2-Like (UBAP2L), which is a central SG node protein in the RNA–protein interaction network of SGs and PBs. UBAP2L binds to the essential SG and PB proteins G3BP and DDX6, respectively. Research on UBAP2L has mostly focused on its role in SGs, but not its connection to PBs. We find that UBAP2L is not solely an SG protein but also localizes to PBs in certain conditions, contributes to PB biogenesis and SG–PB interactions, and can nucleate hybrid granules containing SG and PB components in cells. These findings inform a new model for SG and PB formation in the context of UBAP2L's role.**

## Introduction

Environmental stress challenges cells and organisms to simultaneously reduce energy expenditure and activate stress-protective functions. Translation shutdown, a hallmark of the cellular stress response (Pakos-Zebrucka et al., 2016), lowers metabolic demand and concomitantly frees mRNA, driving the formation of transient cytoplasmic biocondensates, sometimes called membraneless organelles (Riggs et al., 2020). Stress granules (SGs) and processing bodies (PBs) are the most well-studied of such condensates and may contribute adaptive and/or cytoprotective functions to the cellular stress response.

SGs and PBs have been described in diverse taxa including, but not limited to, yeast, humans, and plants (reviewed in Grousl et al., 2022; Kearly et al., 2024; Riggs et al., 2020). Most human cell types and organisms display constitutive PBs that may increase in number and/or size in response to stress (Corbet and Parker, 2019; Kedersha et al., 2005). In contrast, SG formation requires stress or overexpression of key nucleating proteins. Numerous stresses, including heat shock and oxidative stress, induce both SG and PB formation (Riggs et al., 2020). However, some stresses affect SGs and PBs separately. For example, high-dose osmotic stress induces both SG and PB assembly (Kedersha et al., 2016), while low-dose osmotic stress only induces PB formation (Jalihal et al., 2021).

PBs and SGs are distinct entities that differ in size, composition, and mechanism of formation (Riggs et al., 2020). They require and contain some proteins and RNAs in common (Matheny et al., 2019; Ohn et al., 2008; Youn et al., 2018), physically interact under certain conditions (Kedersha et al., 2005; Sanders et al., 2020; Souquere et al., 2009), and may arise from common submicroscopic protein clusters and protein–RNA biophysical interactions (Markmiller et al., 2018; Marmor-Kollet et al., 2020). However, the detailed molecular mechanisms giving rise to SG and PB formation and the significance of their similarities and interactions remain unclear.

Recent work identified the RNA-binding protein (RBP) Ubiquitin Associated Protein 2-Like (UBAP2L) as a key component in SG assembly (Cirillo et al., 2020; Huang et al., 2020; Markmiller et al., 2018; Sanders et al., 2020; Youn et al., 2018) and SG disassembly (Huang et al., 2020). Though UBAP2L also binds to critical PB proteins (Huang et al., 2020; Sanders et al., 2020), its contribution to PB biology remains underexplored. We describe roles for UBAP2L in the formation of PBs as well as SG–PB interaction, which clarify and reframe our understanding of SG and PB biogenesis.

Liquid–liquid phase separation (LLPS), which facilitates the condensation of RNAs and RBPs, drives SG and PB formation (Guillén-Boixet et al., 2020; Molliex et al., 2015; Sanders et al., 2020; Shin and Brangwynne, 2017; Yang et al., 2020). SGs and PBs are both enriched in RBPs containing highly disordered regions, which can enable phase separation by increasing folding and binding capacity in crowded environments (Uversky, 2017; Youn et al., 2018). RNA is required for LLPS to form SGs

[1]Division of Rheumatology, Inflammation and Immunity, Brigham and Women's Hospital, Boston, MA, USA; [2]Department of Medicine, Harvard Medical School, Boston, MA, USA; [3]Department of Biological Sciences, Wellesley College, Wellesley, MA, USA.

Correspondence to Paul Anderson: panderson@rics.bwh.harvard.edu; Pavel Ivanov: pivanov@bwh.harvard.edu.

or PBs (Guillen-Boixet et al., 2020; Sanders et al., 2020; Teixeira et al., 2005); however, the source and identity of the RNA and RBPs differ among these condensates (Matheny et al., 2019).

SGs condense from proteins and mRNAs that accumulate following inhibition of translation initiation, via eIF2α-phosphorylation-dependent or -independent pathways (Hofmann et al., 2021). This converts polysomes into stalled translation preinitiation complexes comprised of polyadenylated (poly(A)$^+$) mRNA, translation initiation factors, and 40S ribosomal subunits (Kedersha et al., 1999; Souquere et al., 2009). These preinitiation complexes are present in SGs (Kedersha et al., 2002) and thus likely serve as the substrate for SG formation via LLPS. RBPs, including translation initiation factors, are essential for consolidating the mRNA freed from translation into SGs and are therefore core SG components (Guillen-Boixet et al., 2020; Kedersha et al., 1999, 2016).

PBs are condensates formed from translationally repressed RNAs (Hubstenberger et al., 2017; Kedersha et al., 2005). RBPs associated with PBs are enriched for proteins involved in mRNA degradation and decay (Decker and Parker, 2012; Hubstenberger et al., 2017; Sheth and Parker, 2003). Accordingly, PBs lack most translation initiation factors, including poly(A)-binding protein (PABP) (Kedersha et al., 2005), yet their induction under stress (formation and/or increase in size and number) still requires polysome disassembly (Andrei et al., 2005). RNA released from translation therefore must associate with RNA degradation proteins to condense into PBs. The mechanism for this transfer of mRNA from translation to degradation is unclear, though eukaryotic translation initiation factor 4E (eIF4E), 4E-T, and DEAD-box helicase 6 (DDX6) likely contribute (Andrei et al., 2005). While stress-induced PB formation and increased PB size depend on translation inhibition (Kedersha et al., 2005; Teixeira et al., 2005), PBs can also be present under optimal conditions with active translation, though the mechanism maintaining them is unclear (Kedersha et al., 2005; Teixeira et al., 2005). High steady-state levels of mRNAs targeted for degradation, possibly due to mRNA exceeding the capacity of translational machinery, might explain the maintenance of constituitive PBs.

RBPs increase the local concentration of RNAs and associated proteins, thereby facilitating LLPS of both SGs and PBs. Recent work indicates that RNA is the primary driver of SG formation (Guillen-Boixet et al., 2020), yet RBPs are essential to facilitate SG formation via LLPS. Though many RBPs are involved in SG condensation, Ras-GTPase-activating protein (SH3 domain) binding protein 1/2 (G3BP) plays particularly important roles under most stress conditions, with the exception of osmotic stress and heat shock (Kedersha et al., 2016). G3BP serves as a scaffold to mediate RNA–protein condensation (Guillen-Boixet et al., 2020; Sanders et al., 2020; Yang et al., 2020). When RNA binds G3BP, G3BP changes conformation, making additional RNA-binding domains (RBDs) accessible and thus enabling LLPS (Guillen-Boixet et al., 2020). G3BP indiscriminately binds RNA (mostly poly(A)$^+$ RNA) to trigger LLPS (Guillen-Boixet et al., 2020; Yang et al., 2020). PB formation, however, does not rely on G3BP. Though G3BP-null cells (G3BP1/2 KO) cannot form SGs, they still form PBs in response to stress (Kedersha et al., 2016;

Sanders et al., 2020). Several proteins contribute to PB formation, including 4E-T and LSM14A (Ayache et al., 2015; Kamenska et al., 2016; Minshall et al., 2009; Ohn et al., 2008); however, the RNA helicase DDX6 appears to be the most critical player, without which cells cannot form PBs in response to sodium arsenite (arsenite) (Ayache et al., 2015; Hubstenberger et al., 2017; Sanders et al., 2020). Importantly, PB formation requires DDX6 repression complexes, rather than other DDX6 complexes, suggesting that the role of DDX6 in translation repression and PB formation are linked (Ayache et al., 2015; Kamenska et al., 2016). Under SG- and PB-inducing stresses, DDX6 is recruited to both SGs and PBs—though it predominantly localizes to PBs (Ayache et al., 2015; Sanders et al., 2020).

Recent proximity-labeling proteomics studies highlighted the interconnected nature of SG and PB proteins and identified additional proteins critical for SG formation (Markmiller et al., 2018; Marmor-Kollet et al., 2020; Youn et al., 2018). Components of both SGs and PBs are near each other, both in the presence and absence of stress (Marmor-Kollet et al., 2020; Youn et al., 2018). Proximity-labeling revealed submicroscopic protein complexes (seeds) containing many known SG proteins, some PB proteins such as enhancers of mRNA-decapping protein 3 (EDC3), and proteins not previously associated with SG or PB biology (Marmor-Kollet et al., 2020). Sanders et al. (2020) described a competitive protein–RNA interaction network integrating SG and PB proteins that regulates SG and PB condensation. UBAP2L was identified in these studies as an important contributor to SG assembly (Markmiller et al., 2018; Sanders et al., 2020; Youn et al., 2018).

Additional studies corroborated the requirement of UBAP2L for canonical SG formation (Cirillo et al., 2020; Huang et al., 2020) and proposed explanations for its mechanistic contribution to SG formation. Sanders et al. (2020) showed that UBAP2L increases G3BP's RNA binding capacity, which is essential for the formation of full-sized canonical SGs. Huang et al. (2020) showed evidence that UBAP2L methylation modulates its association with other SG proteins and subsequently SG assembly. Cirillo et al. (2020) proposed that UBAP2L forms distinct cores upstream of G3BP, which nucleate SGs. However, UBAP2L and G3BP associate with each other independently of stress (Huang et al., 2020; Sanders et al., 2020) and are thus present in constitutive submicroscopic seeds (Marmor-Kollet et al., 2020), suggesting that they contribute to LLPS in concert, rather than sequentially.

UBAP2L is a large RBP with a ubiquitin-associated domain (UBA), an Arginine–Glycine–Glycine domain (RGG), three predicted RBDs, intrinsically disordered regions, and a domain of unknown function (DUF) (Hofmann et al., 2021). The RGG binds several mRNA-bound complexes, as well as rRNA and mRNA in unstressed conditions, and has been reported to be required for SG formation (Huang et al., 2020; Luo et al., 2020). The DUF region is required for G3BP to bind to UBAP2L, facilitates cytoplasmic localization of the protein, and is also important for SG formation (Baumgartner et al., 2013; Huang et al., 2020; Youn et al., 2018). UBAP2L's UBA domain is not required for, though may contribute to, SG formation (Huang et al., 2020; Youn et al., 2018). UBAP2L overexpression induces SGs in WT HeLa cells

(Huang et al., 2020) and partially rescues SGs in U2OS G3BP1/2 KO cells (Sanders et al., 2020).

In addition to a role in SG formation, UBAP2L is proposed to make versatile contributions to biology including translation regulation and survival of UV-induced DNA damage, and is connection to several diseases (Bordeleau et al., 2014; Carlston et al., 2021; Herlihy et al., 2022; Lin et al., 2018; Luo et al., 2020). UBAP2L is upregulated in several cancer tissues (He et al., 2018; Li et al., 2022; Wang et al., 2017) and its depletion has been shown to inhibit cancer cell proliferation (Chai et al., 2016; He et al., 2018; Li and Huang, 2014; Li et al., 2022; Ye et al., 2017; Zhao et al., 2015). Lingerer, *C. elegans'* UBAP2L ortholog, is important for survival and normal embryonic development of offspring (Abbatemarco et al., 2021). Overall, UBAP2L appears to play a significant role in global translation regulation and cell proliferation. The relationship between these functions of UBAP2L has yet to be linked to the stress response or the assembly of SGs and PBs.

Here, we describe new roles for UBAP2L in biocondensates biology. We show that UBAP2L contributes to the formation of not only SGs, but also of PBs, and modulates the interaction between SGs and PBs. We anticipate UBAP2L-mediated SG–PB interaction may be an important part of gene expression regulation in the stress response.

## Results

### UBAP2L modulates the formation and interaction of SGs and PBs

Many key SG proteins have homologs with redundant functions, requiring depletion of both to observe an effect, including G3BP1/G3BP2 (Kedersha et al., 2016; Yang et al., 2020) and T-cell-restricted intracellular antigen-1 (TIA-1)/TIA-R (Gilks et al., 2004). Ubiquitin-associated protein 2 (UBAP2), the homolog of UBAP2L, also localizes to arsenite-induced SGs in U2OS-WT cells (Fig. 1 A). UBAP2 levels are elevated in U2OS UBAP2L KO cells (UBAP2L KO) (Fig. 1 B) as previously reported in MRC5 VA cells (Herlihy et al., 2022), further suggesting that UBAP2 might compensate for UBAP2L. However, siRNA depletion of UBAP2, UBAP2L, and the two together show a deviation from homologous SG proteins. Rather, UBAP2L uniquely contributes to SG formation—as well as to PB formation and association with SGs (Fig. 1, C–E).

Efficient depletion of UBAP2, UBAP2L, or both (Fig. S1, A and B) show UBAP2L contributes to SG and PB formation and to their interaction with each other (Fig. 1, C–E; and Fig. S1 C). Representative images show siUBAP2L and siUBAP2/2L decrease SG formation induced by arsenite, heat shock, or osmotic stress (Fig. 1 C and Fig. S1 C). An hour of 250 µM arsenite induces significantly different numbers of SGs/cell in siUBAP2- and siUBAP2L-treated cells (Fig. 1 D). An additive effect occurs with combined siUBAP2/UBAP2L, depressing SG levels significantly below all controls and siUBAP2 cells (Fig. 1 D). However, though not statistically significant, representative blots show a trend toward decreased UBAP2L levels in siUBAP2-treated cells (Fig. S1 B), suggesting the additive effect on SGs may result from additional depletion of UBAP2L resulting from siUBAP2.

siUBAP2L, but not siUBAP2, also yields smaller SGs (Fig. 1 D). The average number of PBs/cell induced by 250 µM arsenite did not differ significantly among siRNA treatments (Fig. 1 D). However, 100 µM arsenite yielded significantly fewer PBs/cell in siUBAP2L- and siUBAP2/2L-treated cells but not in siUBAP2 cells (Fig. 1 E). UBAP2L depletion in G3BP1/2 KO cells also reduced the percentage of cells forming PBs in response to 100 µM arsenite by ~60% (Fig. S2), further implicating UBAP2L in PB biogenesis. In addition to UBAP2L's unique contribution—distinct from UBAP2—to SG and PB formation, UBAP2L dramatically modulates the association between SGs and PBs. siUBAP2L and siUBAP2/2L reduce the percentage of PBs docking to SGs by over 50%, while siUBAP2 has no effect on docking (Fig. 1 D).

These data introduce UBAP2L as a factor in PB formation and PB interaction with SGs and show that UBAP2 does not functionally compensate for UBAP2L in stress-responsive condensates. UBAP2 recruitment to SGs is unaffected by UBAP2L depletion (Fig. S1 D), indicating a change in UBAP2 localization cannot explain its inability to compensate for UBAP2L. These findings align with the identification of UBAP2L, but not UBAP2, in proximity to key SG proteins prior to stress (Markmiller et al., 2018; Marmor-Kollet et al., 2020).

UBAP2L KO recapitulate the reduction in SG formation, PB formation, and SG–PB docking observed by siUBAP2L-treated cells (Fig. 1, F and G). Reconstituting UBAP2L KO cells with GFP-UBAP2L restored the average percentage of PBs docking to SGs to that of U2OS-WT cells (Fig. 1, F and G). Given the distinct contributions of UBAP2L, the remainder of the study focuses on the biology of UBAP2L, and in particular its robust role in SG–PB interaction.

### UBAP2L is present in PBs in certain conditions

UBAP2L also localizes to PB-like foci in stress conditions that induce PBs, but not SGs. In U2OS-WT cells treated with 0.1 M sorbitol (an osmotic stress) and in G3BP1/2 KO cells treated with 0.1 M sorbitol or arsenite (250 µM), the UBAP2L foci that form coincide with known PB markers: human enhancer of decapping large subunit (HEDLS)/EDC4 (Fig. 2 A), EDC3 (Fig. S3 A) and mRNA-decapping enzyme 1A (DCP1A) (Fig. S3 B) (Ivanov et al., 2019; Kedersha and Anderson, 2007; Kedersha et al., 2005; Kshirsagar and Parker, 2004), proteins involved in RNA decapping. Under these conditions, SGs do not form, as indicated by the homogeneous distribution of eIF3b (a robust SG marker) (Kedersha et al., 2008) throughout the cytoplasm. Previous reports have established that G3BP1/2 KO cells treated with arsenite do not form SGs but do form PBs (Kedersha et al., 2016; Sanders et al., 2020). Representative intensity profile plots reveal coincident intensity peaks for UBAP2L and eIF3b where SGs form (U2OS-WT, arsenite) and coincident intensity peaks for HEDLS and UBAP2L in conditions inducing PBs but not SGs (G3BP1/2 KO, arsenite) (Fig. 2 A). The percentage of SGs or PBs enriched for UBAP2L varies considerably with granule identity and the cellular context (Fig. 2 B). Nearly all arsenite-induced SGs in WT cells and arsenite-induced PBs in G3BP1/2 KO cells contain UBAP2L (Fig. 2 B). About 45% of PBs that form in U2OS-WT cells—which also have SGs—contain UBAP2L. However,

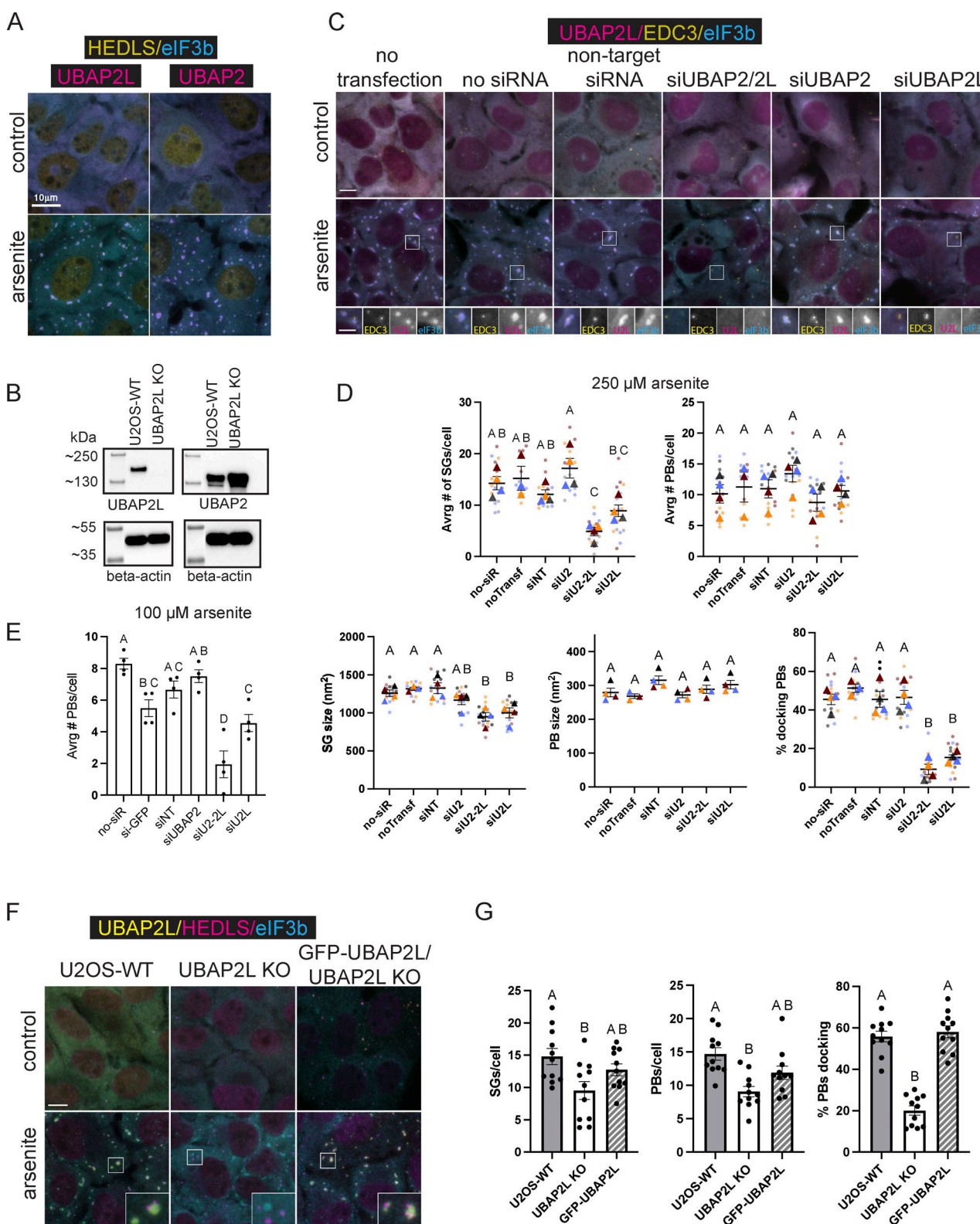

**Figure 1.  UBAP2L contributes to the formation of SGs and PBs and their interaction. (A)** UBAP2L and its homolog, UBAP2, localize to canonical arsenite-induced SGs with docked PBs in U2OS-WT cells. Representative images show immunostaining of U2OS-WT cells untreated and treated with arsenite (250 μM, 60 min). Magenta = UBAP2L or UBAP2; cyan = eukaryotic translation initiation factor 3 subunit B (eIF3b) (SG marker); yellow = human enhancer of decapping large subunit (HEDLS)/EDC4 (PB marker). Scale bar = 10 μm. **(B)** UBAP2 expression levels are elevated in UBAP2L KO cells. **(C)** Representative images show U2OS-WT cells treated with siRNA against UBAP2L, UBAP2, UBAP2, and UBAP2L together (UBAP2/2L), or a non-targeting siRNA and followed by arsenite treatment (250 μM, 60 min). Magenta = UBAP2L; cyan = eIF3b (SG marker); yellow = EDC3 (PB marker). Scale bar = 10 μm in main figures and 5 μm in insets. **(D)** Quantification of SGs and PBs in siRNA-treated cells stressed with 250 μM arsenite, 60 min. Plots show the average number and size of SGs and PBs/cell,

and the % of PBs docking to SGs. Each dot represents the average of the cells visible in one image. Four images were taken for each of four biological replicates (independent experiments). Large triangles represent the mean results from the images ($n = 4$ technical replicates) of one biological replicate. Triangles and circles in different shades of the same color are from the same independent experiment. Error bars show standard error of the mean. Means of biological replicates ($n = 4$) were compared to each other with a one-way ANOVA and a Tukey post-hoc test. Treatments with different letters indicate statistical difference at the level of $P < 0.05$. **(E)** Quantification of PBs in siRNA-treated cells stressed with 100 µM arsenite (60 min) shows siUBAP2L significantly reduces the average number of PBs per cell. Each data point represents the average calculated from cells in one image. $N = 4$ images (technical replicates) analyzed for each treatment of one independent experiment. Data are plotted as the mean of images analyzed with standard error of the mean. Treatments with different letters indicate statistical significance at the level of $P < 0.05$ determined by a one-way ANOVA with a Tukey post-hoc test. **(F)** Representative immuno-fluorescence images of U2OS-WT, UBAP2L CRISPR KO, and UBAP2L KO cells reconstituted with UBAP2L (GFP-UBAP2L/UBAP2L KO), untreated (control) or treated with arsenite (250 µM, 60 min) and immunostained for SG and PB markers. Cyan = eIF3b (SG marker); magenta = HEDLS (PB marker); yellow = UBAP2L (in U2OS-WT cells) and GFP-UBAP2L (in GFP-UBAP2L/UBAP2L KO cells). SGs appear blue or light green (positive for eIF3b in UBAP2L KO cells or double positive for eIF3b and UBAP2L in WT and reconstituted cells) and PBs are magenta (positive for HEDLS). Scale bar main images = 10 µm, inset scale bar = 5 µm. **(G)** Quantification of the average number of SGs/cell, PBs/cell, and average % of total PBs docking to SG in U2OS-WT, UBAP2L KO, and GFP-UBAP2L cells treated with arsenite (shown in F). Means for each cell type are presented with standard error of the mean and compared by a one-way ANOVA. Cell types annotated with different letters are significantly different from each other ($P < 0.05$). Each data point graphed represents the average from cells in one image. Three independent experiments were performed and three to four images (technical replicates) per experiment were analyzed. Source data are available for this figure: SourceData F1.

this is likely due to the PBs docking to SGs, resulting in close proximity that may allow for transfer of UBAP2L to PBs—or simply the detection of visually overlapping signal. Of the PBs that are non-docking (not in contact with a SG), only ~7% of these contain UBAP2L (Fig. 2 B). Constitutive PBs that are present in the absence of stress (and therefore also not docking, as there are no SGs present) lack UBAP2L (Fig. 2 B), suggesting that stress is required for UBAP2L to condense into SGs or PBs. Comparing PBs in these different contexts, a relationship to stress and SG formation emerges: UBAP2L recruitment to PBs depends on stress, but is attenuated by the presence of SGs (Fig. 2, A and B).

Recovery from arsenite also induces UBAP2L recruitment to PBs. Representative images of cells recovering from arsenite treatment show disassembly of SGs and PBs and capture a transition in the cells from displaying stress-induced SGs and PBs to just PBs. Recovery effectively achieves mild stress, where PBs are present but not SGs, since SGs disassemble first. SGs disassemble by ~60 min recovery (Fig. 2 C). We observe relocalization of UBAP2L from SGs to PBs during this time course (Fig. 2 C). Immediately following 60 min of 250 µM arsenite, intensity plots show coinciding peaks for UBAP2L and eIF3b (representing SGs) and separate HEDLS peaks (representing PBs). After 30 min recovery, the SG peaks are smaller, and by 60 min recovery most SGs are absent and UBAP2L appears in PBs (indicated by white arrow). The presence of UBAP2L in PBs is still observed at 90 min recovery and is less prominent at 120 min recovery (Fig. 2 C). Together, these data reveal UBAP2L is preferentially recruited to SGs, but will localize to PBs under conditions lacking a robust SG response.

## UBAP2L "cores" in arsenite-treated G3BP1/2 KO cells are bona fide PBs

Given that UBAP2L has thus far only been documented as an SG protein, we further characterized the PB-like UBAP2L-positive foci that form in G3BP1/2 KO cells treated with arsenite (Fig. 2, Fig. S3, and Fig. 3, A and B). Such foci were previously referred to as UBAP2L "cores" and proposed to serve as precursors to SGs (Cirillo et al., 2020). The diversity of stress-induced phase-separated RNP (ribonucleoprotein) granules and precursors

(Marmor-Kollet et al., 2020; Riggs and Ivanov, 2022) warranted further characterization to confirm their identity. PB-specific proteins localize to UBAP2L-positive foci induced by arsenite in G3BP1/2 KO cells, while SG-specific proteins are not robustly recruited (Fig. 3, A and B). Known PB components (DCP1a, HEDLS, and XRN1) displayed ~20% of their total protein signal in the UBAP2L foci, when normalized to total PB area, in G3BP1/2 KO cells (Fig. 3 B). On the contrary, SG proteins displayed on average ~2% of their total signal in these foci (Fig. 3 B). Even fragile X mental retardation autosomal homolog 1 (FXR1), a UBAP2L binding partner and SG protein (Huang et al., 2020; Kedersha et al., 2005), was not robustly recruited to these UBAP2L-positive foci (Fig. 3 B), suggesting they are not UBAP2L-centric foci. About 12% of UBAP2L signal localized to these UBAP2L-positive foci, which was not significantly different from other dual localizing SG/PB proteins, but did differ significantly from recruitment of known SG proteins (Fig. 3 B), fitting our understanding of the protein. Pre-stress seed proteins FXR1, TIA1, and DAZAP1 (Marmor-Kollet et al., 2020) were not robustly recruited to the UBAP2L-positive foci, indicating that the foci are not pre-stress seeds (Fig. 3 B, indicated with asterisk). Neither poly(A)⁺ mRNA nor PABP were observed in these UBAP2L-positive granules (Fig. 3 C), consistent with previous in situ hybridization experiments in which PBs lacked visible poly(A)⁺ mRNA (Cirillo et al., 2020; Kedersha et al., 2005). While transcriptomic analysis reveals a complex RNA landscape in PBs and SGs (Hubstenberger et al., 2017), the lack of robust poly(A)⁺ mRNA signal contributes to our results indicating these UBAP2L-positive foci in G3BP1/2 KO cells are PBs.

Furthermore, UBAP2L foci dynamics are consistent with stress-induced condensate biology, including that of PBs. G3BP1/2 KO cells treated simultaneously with arsenite and cycloheximide (which stabilizes polysomes [Dmitriev et al., 2020]), do not form UBAP2L-positive granules (Fig. 3 D; Cirillo et al., 2020), indicating that their formation depends on polysome disassembly. This is consistent with PB biology, as cycloheximide treatment prevents their stress-induced assembly (Andrei et al., 2005). While SG formation also depends on polysome disassembly, this result provides further evidence that these foci are not pre-stress seeds (Fig. 3 B, indicated with an asterisk), whose

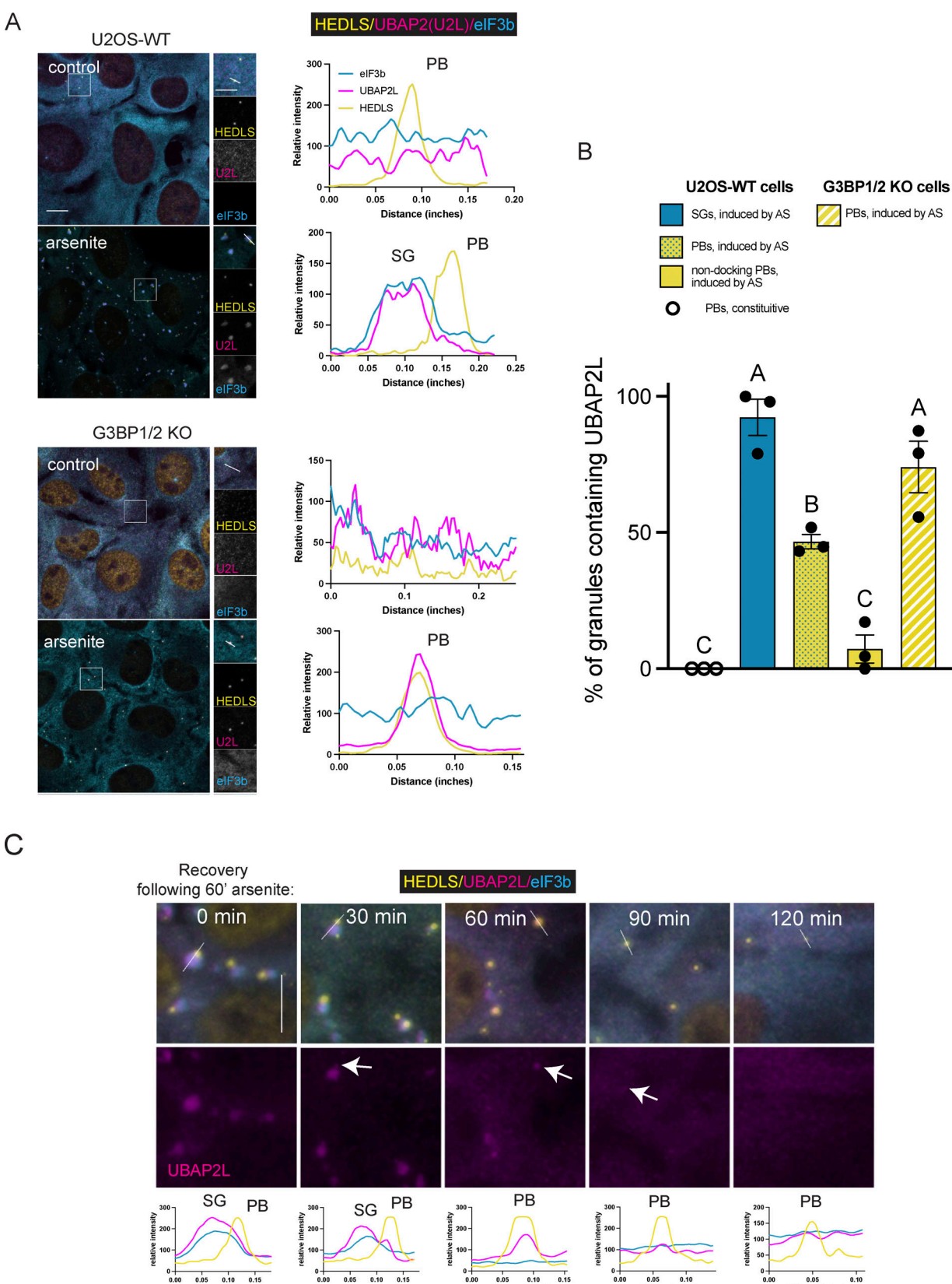

Figure 2. **UBAP2L subcellular localization depends on stress conditions.** UBAP2L is recruited to PBs during stresses that do not induce SGs but still induce PB formation. **(A)** Representative confocal images of U2OS-WT and G3BP1/2 KO cells untreated (control) or treated with arsenite (AS) (250 μM, 60 min) and immunostained for UBAP2L and SG and PB markers. Images shown are a sum of the signal from a 12-slice z-stack, imaged with consistent parameters for all treatments and cell types. Representative intensity profile plots shown to the right of each panel correspond with the white line drawn in the merged expanded

panel. 250 µM arsenite induces SGs and PBs in U2OS-WT cells, but only PBs in G3BP1/2 KO cells. Cyan = eIF3b (SG marker); magenta = UBAP2L; yellow = HEDLS (PB marker). Experiments performed with EDC3 and DCP1a also show UBAP2L recruitment to PBs induced by arsenite, as well as osmotic stress (Fig. S3). Scale bar = 10 µm in main images and 5 µm in insets. Intensity profile plots through SGs and PBs show enrichment of UBAP2L in SGs in U2OS-WT cells treated with arsenite, and enrichment of UBAP2L in PBs in G3BP1/2 KO cells treated with arsenite, which induces PBs but not SGs. **(B)** Quantification of the granules containing UBAP2L using confocal images, shown in A. SGs and PBs were counted as UBAP2L-positive if UBAP2L mean intensity in the granules was at least 3× that of its mean intensity in the cytoplasm (excluding SGs or PBs). Quantification was performed on z-stack summed confocal images on three fields within each sample. Each data point plotted represents the percentage of granules of interest containing UBAP2L from a single image. Mean values from images are plotted showing the standard error of the mean for each granule type. Means were compared to each other with a one-way ANOVA followed by a Tukey's post-hoc test. Granule types annotated with different letters represent a significant difference ($P < 0.05$) in UBAP2L containing granules. Blue bar = arsenite-induced SGs in U2OS-WT cells, yellow bars with blue dots = all arsenite-induced PBs in U2OS-WT cells (docking and non-docking), solid yellow bars = non-docking PBs in U2OS-WT cells treated with arsenite, and striped yellow and white bars = PBs induced by arsenite in G3BP1/2 KO cells. **(C)** UBAP2L is recruited to PBs during recovery from arsenite stress. Representative immunostained images are shown for U2OS-WT cells treated with arsenite (250 µM, 60 min) and allowed to recover in regular DMEM media (used for maintaining the cells) for 0, 30, 60, 90, or 120 min. Representative intensity profile plots reveal a shift in UBAP2L from SGs, aligning with the eIF3b peak, to PBs during recovery, aligning with the HEDLS peak. Intensity profiles correspond to white lines shown on the top panel of microscopy images. White arrows in bottom panel of microscopy images point to UBAP2L signal in PBs following recovery from arsenite treatment. In images and intensity plots, cyan = eIF3b (SG marker); magenta = UBAP2L; yellow = HEDLS (PB marker).

formation does not require polysome disassembly (Marmor-Kollet et al., 2020). Considered together with the protein localization data (Fig. 2, Fig. S3, and Fig. 3 A), we conclude that the UBAP2L-positive foci induced by arsenite in G3BP1/2 KO cells are bona fide PBs.

### UBAP2L modulates SG–PB association

Arsenite induces the formation of SGs with transiently docking PBs in U2OS-WT cells (Fig. 1, C–F) (Kedersha et al., 2005; Sanders et al., 2020). Reconstitution of UBAP2L KO cells with varying levels of UBAP2L using a tet-inducible (doxycycline [dox]-inducible) system shows that UBAP2L expression alters the association of PBs and SGs in a dose-dependent manner (Fig. 4, A–C). Representative images show a shift from distinct PBs and SGs to overlapping signals that are barely distinguishable (Fig. 4, A and B). Expanded images and their quantification (Fig. 4, B and D) show changes in SG and PB size, number, and relation to each other with increasing levels of UBAP2L.

5 h dox induction of UBAP2L in KO cells, followed by arsenite, restores formation of SGs with distinct docking PBs (Fig. 4 B). Longer induction of UBAP2L (24 h dox treatment) followed by arsenite treatment yields large granules containing SG and PB proteins (Fig. 4, A and B). As UBAP2L levels increase, the average number of PBs/cell increases while the average PB size decreases (Fig. 4 D). The increase in PB number is consistent with the observation that UBAP2L KO cells form fewer PBs (Fig. 1), showing a role for UBAP2L in PB biogenesis. The smaller PB size, however, may be a result of their increased interaction with SGs. Comparing UBAP2L KO cells (no dox) and cells overexpressing UBAP2L (24 h dox) shows that overexpression of UBAP2L significantly increases the mean intensity of HEDLS in the area defined by the SGs (Fig. 4 D). Additionally, substantially more of the PB-occupied area overlaps with SG occupied area when UBAP2L is overexpressed. The SG area overlapped with nearly 40% of the PB signal when UBAP2L was overexpressed versus ~5% overlap in UBAP2L KO cells (Fig. 4 D). These two analyses and the representative images show UBAP2L overexpression increases the association between PBs and SGs, causing closely docking PBs and/or forming a hybrid granule in which distinct PB-like foci surround or may even be contained within a large granule (Fig. 4, A–D).

The LALA mutant form of UBAP2L, which does not bind G3BP1 (Baumgartner et al., 2013; Youn et al., 2018), displays the same effect as WT UBAP2L (Fig. S4 A), indicating that UBAP2L-mediated SG–PB interaction and formation of hybrid granules does not strictly require UBAP2L:G3BP binding. In addition, hybrid granules nucleated by UBAP2L overexpression and arsenite do not require key SG proteins, G3BP and Fragile X proteins, which were shown to interact with UBAP2L (Huang et al., 2020; Sanders et al., 2020). G3BP1/2 KO and FMR1/FXR1/FXR2 triple-KO U2OS cells (FFF KO) still form hybrid granules induced by UBAP2L overexpression (Fig. 4 E), indicating that neither G3BP nor Fragile X-related proteins alone is essential for UBAP2L to nucleate hybrid granules. However, it is possible that either G3BP or Fragile X proteins must be expressed to support the formation of hybrid granules and that simultaneous depletion of the two proteins might interfere with UBAP2L's activity. Importantly, the hybrid granule phenomenon is not induced by all SG proteins. Neither FXR1 overexpression in FFF KO cells nor G3BP1 overexpression in G3BP1/2 KO cells followed by arsenite treatment yields the hybrid granules characteristic of UBAP2L overexpression (Fig. 4 F). In the case of G3BP1, DDX6 predominately localizes to PBs docking at SGs rather than integrating with the SG (Fig. 4 F). This suggests that the hybrid granule phenomenon is a unique property of UBAP2L overexpression among SG-nucleating proteins (Fig. 4 F).

To confirm that the changes to HEDLS and DDX6 foci relative to SGs represented a change in PBs themselves, not just in certain proteins, we further examined the composition of the hybrid granules nucleated by UBAP2L. Numerous SG, PB, and dual localizing proteins are visible in the hybrid granules (Fig. S4 B). Many PB proteins appear in distinct foci within and/or surrounding the UBAP2L-positive granules (DCP1A, DDX6, HEDLS) while for some proteins (eIF4E and 4-ET), there is no clear morphological separation between SGs and PBs (Fig. S4 B). Proteins with homogenous distribution are indicative of a single-phase granule (Sanders et al., 2020) and reflect protein redistribution as a result of UBAP2L abundance, suggesting increased mixing and interaction among the proteins present.

In the absence of arsenite (Fig. S4 B), UBAP2L overexpression nucleates smaller granules containing PB-specific components (e.g., HEDLS and DCP1A), shared components which predominantly

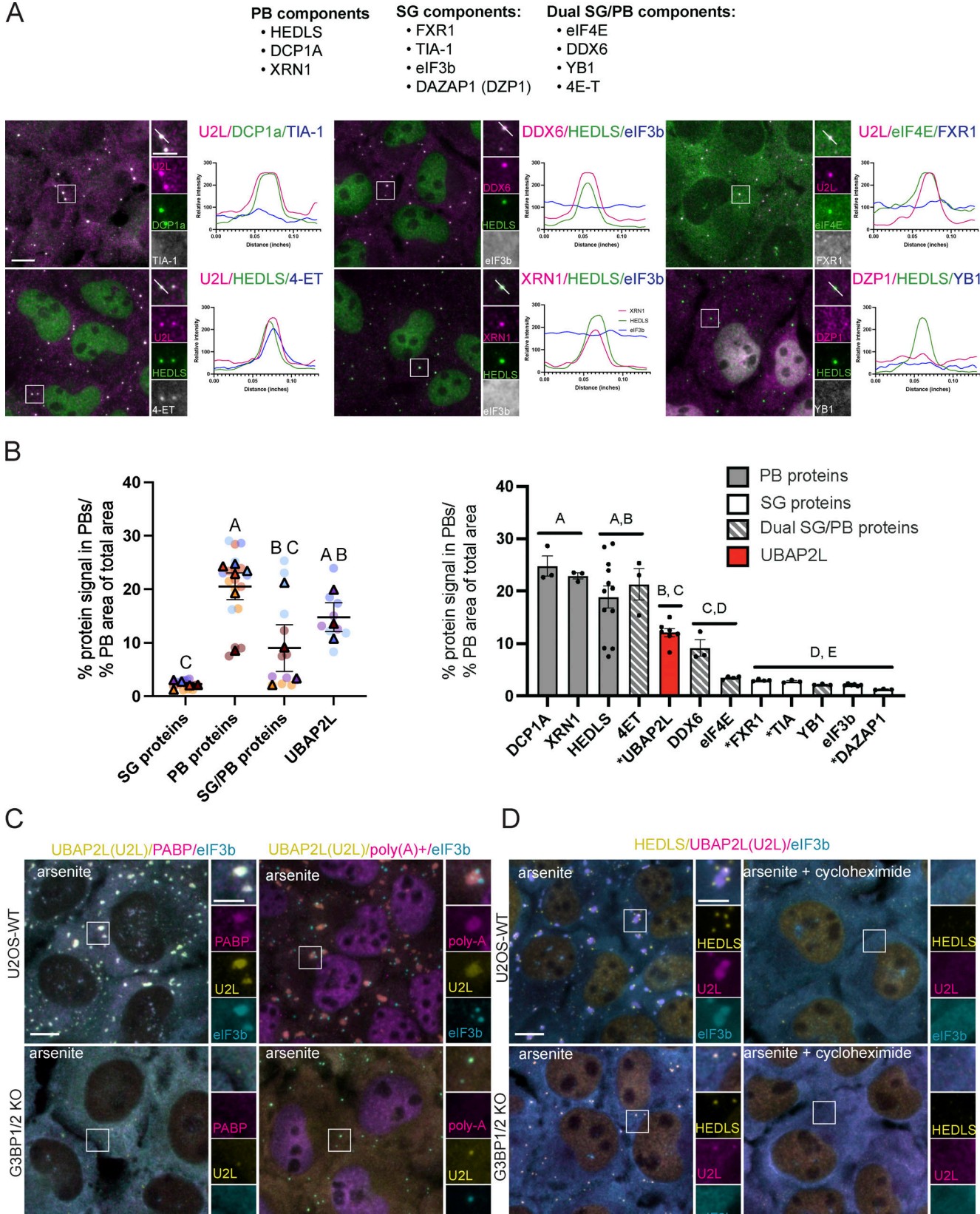

Figure 3. **UBAP2L-positive foci induced by arsenite in G3BP1/2 KO cells are bona fide PBs. (A)** Representative immunostained images and intensity profile plots show coinciding signal for UBAP2L foci and PBs in arsenite-treated (250 μM, 60 min) G3BP1/2 KO cells. Profile plots show the relative signal intensity through the white line in the merged inset to the left of the plot. Known PB proteins, SG proteins, and dual localizing components are included for comparison. Scale bars = 10 μm in main images and 5 μm in insets. **(B)** Quantification of protein recruitment to UBAP2L-positive foci (representative images

shown in A). Data are grouped by protein category (left) and separated by individual proteins (right). In the grouped data, the mean of three to six biological replicates for each protein category is plotted with the standard error of the mean. Each triangle represents the mean measurements from three images from one independent experiment. The associated replicates are shown in circles in the same color as the triangle, but in a lighter shade. Means of independent experiments were compared to each other with a one-way ANOVA followed with a Tukey post-hoc test. Protein categories displaying no letters in common differ statistically (P < 0.05) in their recruitment to PBs. The graph on the right shows the data separated by individual protein. Gray bars = PB proteins; white bars = SG proteins; gray and white striped bars = dual SG & PB localizing proteins. UBAP2L is shown in red. Asterisks indicate proteins identified in pre-stress seeds (Marmor-Kollet et al., 2020). Data points within a protein represent measurements from at least three images from one independent experiment. For some proteins (HEDLS and UBAP2L) measurements from images from multiple independent experiments are shown. This is due to use of the some of the same antibodies on more than one sample. Mean signal in PBs normalized to the PB area in each image was compared among proteins with a one-way ANOVA with a Tukey post-hoc test. Proteins annotated with different letters are statistically different from each other (P < 0.05) in their recruitment to PBs. **(C)** PABP (immunostained) and poly(A)$^+$ RNAs (fluorescence in situ hybridization) localize to SGs in U2OS-WT cells treated with arsenite (250 µM, 60 min), but are not recruited to UBAP2L-positive granules in arsenite-treated G3BP1/2 KO cells. Yellow = UBAP2L; magenta = PABP or poly(A)$^+$ mRNA; cyan = eIF3b (SG marker). Scale bars = 10 µm in main images and 5 µm in insets. **(D)** UBAP2L-positive granules exhibit PB behavior and dynamics. U2OS-WT and G3BP1/2 KO cells were treated with arsenite (250 µM, 60 min) or arsenite (250 µM, 60 min) + cycloheximide (50 µg/ml) simultaneously. Cycloheximide, a translation elongation inhibitor, stabilizes polysomes, and blocks SG and PB formation. Yellow = HEDLS (PB marker); magenta = UBAP2L; cyan = eIF3b (SG marker). Scale bars = 10 µm in main images and 5 µm in insets.

localize to PBs in arsenite-treated U2OS-WT cells (e.g., 4E-T and DDX6), and many SG-specific proteins (Fig. S4 B). In particular, known UBAP2L binding partners (G3BP, FXR1, and DDX6) (Huang et al., 2020; Sanders et al., 2020), as well as other RBPs (Y-box binding protein 1 [YB1] and TIA-1), are heavily recruited. Two translation initiation factors examined (eIF3b and eIF4G) require arsenite stress for their recruitment (Fig. S4 B).

Cells overexpressing UBAP2L, without any arsenite treatment, have small spherical granules containing UBAP2L and HEDLS, but lacking eIF3b (Fig. S4, B and C). These granules are PB-like and contain UBAP2L; however, they appear larger on average than typical PBs. After 60 min of 100 µM arsenite treatment, eIF3b joins these granules, and they increase in size and become more irregularly shaped. HEDLS appears in smaller puncta, largely overlapping with the same area as the UBAP2L-positive granules. Therefore, rather than SGs and PBs forming independently and then coming together into hybrid granules, PB-like UBAP2L-positive foci form and then recruit SG proteins, developing into hybrid granules (Fig. S4 C).

In U2OS-WT cells, a similar progression occurs. However, UBAP2L is not visible in PBs but condenses into granules at the same time as eIF3b, after about 45 min of arsenite treatment. Interestingly, SGs appear to start forming very near to or directly overlapping with PBs (Fig. S4 C), before separating into distinct SGs with docking PBs after ~45–60 min arsenite (Fig. S4 C). In UBAP2L KO cells, PB and SG formation is delayed compared with U2OS-WT cells, and SGs form near PBs, resulting in little docking. These data suggest an important role for UBAP2L in SG formation near PBs, leading to classic docking PBs.

UBAP2L is an RBP comprised of disordered regions, RNA binding motifs, and protein binding domains (Hofmann et al., 2021) (Fig. 5 A). Formation of SGs requires UBAP2L's RGG region, which binds RNA and protein, and its DUF domain, which binds the essential SG protein G3BP1 (Baumgartner et al., 2013; Huang et al., 2020; Luo et al., 2020; Youn et al., 2018). The RGG and DUF regions are highly conserved between UBAP2L and UBAP2 (Fig. S5 A); however, neither is required for SG:PB docking (Fig. S5 B). Expression of dRGG-UBAP2L or dDUF-UBAP2L in UBAP2L KO cells restores docking, although dRGG-UBAP2L does not result in hybrid granules, but rather canonical

SGs with docked PBs. dDUF-UBAP2L nucleates hybrid granules resembling those formed by overexpression of full-length UBAP2L (Fig. S5 B). Arsenite treatment of cells transiently expressing progressively shorter UBAP2L fragments indicates amino acids 205–290 are essential for SG:PB docking and for the formation of hybrid granules (Fig. 5, B and C). Cells expressing full-length UBAP2L, amino acids 91–1,087, or 205–1,087 form significantly more hybrid SGs compared with cells expressing amino acids 291–1,087 or 495–1,087 (Fig. 5, B and C). The dramatic shift from hybrid granule formation to lack thereof occurs when amino acids 205–290 are no longer included, indicating the region is critical in hybrid granule formation. 205–290 contains two predicted RBDs (Fig. 5 A) (Hofmann et al., 2021), suggesting that RNA binding may be essential for SG:PB docking and condensation into hybrid granules. Furthermore, amino acids 205–290 are not well conserved between UBAP2L and UBAP2 (Fig. S5 A), which is consistent with the ability of UBAP2L, but not UBAP2, to mediate docking (Fig. 1 D). Interestingly, expression of the DUF domain alone actually blocks the formation of SGs, but not PBs, in arsenite-treated UBAP2L KO cells (Fig. S5 B). When only UBAP2L's RGG domain is expressed, it is constrained to the nucleus, consistent with the DUF domain being required for for cytoplasmic localization (Huang et al., 2020).

## Discussion

UBAP2L is an RBP centrally located within the protein interaction network common to both SGs and PBs (Marmor-Kollet et al., 2020; Sanders et al., 2020). Our results corroborate the documented role of UBAP2L in SG formation (Cirillo et al., 2020; Huang et al., 2020; Markmiller et al., 2018; Sanders et al., 2020; Youn et al., 2018) and expand UBAP2L's role to include PB biology (Fig. 1). We find that UBAP2L contributes to the formation of PBs, is recruited to stress-induced PBs when SG formation is suppressed, mediates SG:PB docking, and nucleates hybrid granules containing both SG and PB proteins. These findings are consistent with the protein–RNA network model of biocondensate formation (Guillen-Boixet et al., 2020; Sanders et al., 2020) and clarify UBAP2L's role within this dynamic network (Fig. 6).

Figure 4.    **UBAP2L regulates SG–PB association under stress independently of G3BP or Fragile X-related proteins.** Tet-inducible GFP-UBAP2L-WT/ UBAP2L KO (GFP-UBAP2L) and GFP-UBAP2L-LALA/UBAP2L KO (GFP-LALA) U2OS cells were treated with 100 ng/ml dox for 0, 5, 8, and 24 h to induce varying levels of UBAP2L-WT or UBAP2L-LALA (Fig. S4 A) prior to treatment with arsenite (250 µM, 60 min). LALA is a mutant version of UBAP2L unable to bind G3BP1

(Baumgartner et al., 2013; Youn et al., 2018). **(A)** Following dox treatment for 0, 5, 8, or 24 h to induce UBAP2L, cells were treated with arsenite (250 µM, 60 min) prior to immunostaining. Magenta = eIF3b (SG marker); cyan = HEDLS (PB marker). Scale bar = 20 µm. **(B)** Insets corresponding to white boxes in A show representative granules depicting the changing nature and relationship between SGs and PBs when UBAP2L is overexpressed. Magenta = eIF3b (SG marker); cyan = HEDLS (PB marker); green = GFP-UBAP2L. Scale bar = 5 µm. **(C)** Western blot analysis corresponding to samples shown in A and B shows UBAP2L induction. **(D)** UBAP2L overexpression increases PB number, decreases PB size, increases HEDLS signal in SGs, and increases the spatial overlap between SGs and PBs. Graphs show quantification of the average number of PBs and the average size of PBs per cell in cells with UBAP2L expression ranging from none (0 h dox) to high overexpression (24 h dox). Data are represented as the mean of three technical replicates (images analyzed) within one independent experiment. Error bars show the standard error of the mean. Only PBs 2–50 pixels in area were considered in the analysis to exclude large condensations of PB proteins (coinciding with SG signal) that do not resemble PBs. HEDLS intensity was measured inside and outside of SGs in cells without or overexpressing UBAP2L. The spatial overlap in SG and PB area was also compared in cells without or overexpressing UBAP2L. Analysis was performed on three images for each sample. Mean PB size and number were compared with a one-way ANOVA followed by a Tukey post-hoc test. Treatments without any letters in common are significantly different (P < 0.05). Mean HEDLS intensity in the SGs and % PB area occluded by SGs were compared with an unpaired two-tailed $t$ test. ** indicates P = 0.0043; *** indicates P = 0.0002. **(E)** UBAP2L overexpression condensate formation does not require G3BP or Fragile X-related proteins. UBAP2L is overexpressed in UBAP2L KO, G3BP1/2 KO, and FMR1/FXR1/FXR2-3KO U2OS (FFF KO) cells via tet-inducible cell lines: tet-on-GFP-UBAP2L/UBAP2L KO; tet-on-GFP-UBAP2L in G3BP1/2 KO; tet-on-GFP-UBAP2L in FFF KO. All cells were treated with dox (100 ng/ml) for 24 h to induce UBAP2L overexpression prior to arsenite treatment (250 µM, 60 min). Green = GFP-UBAP2L; cyan = HEDLS (PB marker); magenta = eIF3b (SG marker). Scale bar = 10 µm in main figures and 5 µm in insets. **(F)** Overexpression of known UBAP2L binding partners does not mimic the UBAP2L overexpression phenomenon. UBAP2L, G3BP, and FXR1 were overexpressed via tet-inducible systems in UBAP2L KO, G3BP1/2 KO, and FFF KO cells, respectively, with dox treatment (100 ng/ml, 24 h) prior to arsenite treatment (250 µM, 60 min). UBAP2L and FXR1 are GFP-tagged in UBAP2L KO and FFF KO cells, respectively, while G3BP is APEX-tagged in G3BP1/2 KO and thus detected by a G3BP1 antibody. Green = GFP-UBAP2L, G3BP1, or GFP-FXR1; magenta = DDX6 (shared SG and PB protein, with predominant PB localization); cyan = eIF3b (SG marker). Scale bar = 10 µm in main figures and 5 µm in insets. Source data are available for this figure: SourceData F4.

Under optimal conditions, UBAP2L associates with polysomes and modulates translation (Huang et al., 2020; Luo et al., 2020). It is also present in submicroscopic protein "seed" complexes containing SG and PB proteins (Marmor-Kollet et al., 2020), and constitutes a central node in the RNA–protein interaction network (Sanders et al., 2020), associating both with the SG-specific protein G3BP1 and the PB-essential protein DDX6 (Ayache et al., 2015; Kedersha et al., 2016; Sanders et al., 2020). As a large RBP with multiple RBDs, SG- and PB–protein associations, and interaction with translation machinery, UBAP2L is poised to mediate the stress-responsive condensation of RNA and protein upon translational arrest.

UBAP2L depletion reduces the average number of SGs and PBs/cell by ~40% (Fig. 1). UBAP2L depletion effects are not as dramatic as those of G3BP or DDX6 KO, as arsenite-treated G3BP1/2 KO cells do not form SGs (Kedersha et al., 2016), and DDX6 KO cells form, on average, <1 PB per cell (Ayache et al., 2015). However, UBAP2L depletion also reduces the size of SGs (Fig. 1). More intriguing, perhaps, than UBAP2L's role in SG and PB formation, is its contribution to interactions between SGs and PBs. SG:PB docking decreases ~50% in UBAP2L KO cells and is restored to WT docking levels when UBAP2L is reconstituted (Fig. 1 F).

We propose that UBAP2L acts as a bridge, pulling SGs and PBs near to each other via protein–protein and protein–RNA interactions, some of which are mediated by RNA released from translation. This hypothesis is supported by our data showing that UBAP2L overexpression nucleates hybrid granules comprised of both SG and PB proteins (Fig. 4 and Fig. S5). Overexpression of the *C. elegans* ortholog of UBAP2L (Lingerer) also forms cytoplasmic Lingerer-positive puncta containing SG and PB proteins (Baumgartner et al., 2013), revealing the conservation of this phenomenon across species. Importantly, the ability of UBAP2L overexpression to mediate RNA and protein condensation into hybrid granules is not a generic property of SG or PB proteins (Fig. 4 F). Rather it is a function of UBAP2L's position in the SG–PB network in which altered abundance of any single

protein changes the relative stoichiometries and thus shifts the system in one direction or another (Guillen-Boixet et al., 2020; Sanders et al., 2020; Yang et al., 2020).

UBAP2L's ability to modulate SG:PB docking and nucleate hybrid granules is likely due to both its protein binding partners within the SG–PB network (Sanders et al., 2020) and its intrinsic RNA binding capacity. Docking and hybrid granule formation require amino acids 205–290, which contains two predicted RBDs (Fig. 5) and a region (amino acids 239–290) reported to facilitate association with FXR2 and PABPC1 (Huang et al., 2020). The RGG domain, which binds RNA, modulates translation, and associates with other SG proteins (Huang et al., 2020; Luo et al., 2020; Youn et al., 2018), is not required for SG:PB docking; however, it is critical for hybrid granules to form (Fig. S5 B). Both region 205–290 and the RGG domain likely contribute to RNA and protein binding capacity critical for docking, though we suspect that the RGG is especially important for increasing RNA binding to nucleate hybrid granules. Since mRNAs can be bound by SG and PB proteins, RNA binding is a realistic mechanism by which UBAP2L may condense SG and PB components into single phase granules. RGG mediates UBAP2L and RNA interaction (Luo et al., 2020) and is sufficient to nucleate SGs in G3BP1/2 KO cells (Sanders et al., 2020). Furthermore, UBAP2L predominantly binds to the coding region of mRNA (Luo et al., 2020), thus it likely binds mRNA regardless of the adenylation and capping state. Therefore, mRNAs freed from stress-induced translation inhibition may be bound by UBAP2L, whether they are targeted for degradation or maintained in preinitiation complexes. In the presence of excess UBAP2L, we expect that it can bind more free RNA, pulling together RNAs and proteins from translation and degradation pathways and condensing hybrid granules into being.

Overexpression of other RBPs, such as tristetraprolin (TTP)/ ZFP36, Bromodomain and PHD finger-containing protein 1 (BRF1), and cytoplasmic polyadenylation element-binding protein 1 (CPEB), also alters SG–PB interaction and fusion (Kedersha et al., 2005; Stoecklin and Kedersha, 2013;

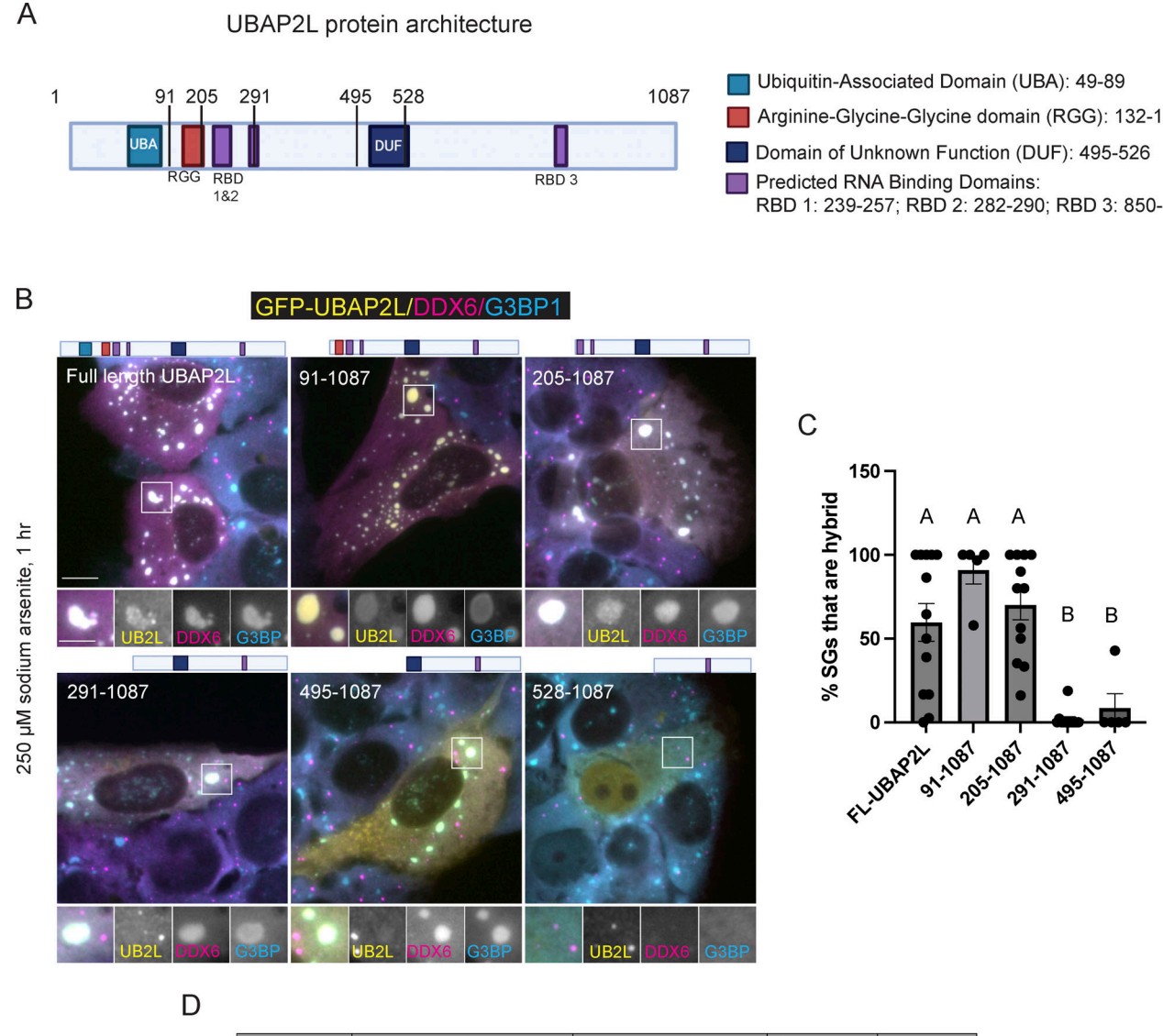

Figure 5. **UBAP2L region containing predicted RBDs is essential for SG:PB docking and formation of hybrid granules.** UBAP2L KO cells transiently expressing UBAP2L fragments reveal amino acids 205–291 are required for the formation of hybrid granules. **(A)** UBAP2L protein architecture, demarcating domains of interest. **(B)** UBAP2L KO cells transiently transfected with UBAP2L fragments and subsequently treated with arsenite (250 μM, 60 min) reveal the protein region required for SG–PB interaction. Yellow = GFP-UBAP2L protein fragment as indicated; magenta = DDX6 (predominately in PBs and weakly recruited to SGs); cyan = G3BP1. Scale bar = 10 μm in main image and 5 μm in insets. **(C)** Quantification of hybrid granule formation in cells expressing different UBAP2L constructs. Each data point represents analysis of one cell. Full-length UBAP2L, n = 13; 91–1,087, n = 5; 205–1,087, n = 12; 291–1,087, n = 12, 495–1,087,

*n* = 5. Mean % hybrid SGs were compared across UBAP2L constructs with a one-way ANOVA with a Tukey's post-hoc multiple comparisons test. Constructs with different letters differ significantly (P < 0.05) from each other in hybrid granule formation. **(D)** Summary of UBAP2L fragment architecture, biocondensate formation, and subcellular localization. Comparison of expression of UBAP2L amino acids 205–1087 versus 291–1087 reveals that amino acids 205–291, a region containing two predicted RBD, is essential for the association of PBs with SGs.

Wilczynska et al., 2005). Each of these proteins, like UBAP2L, exhibits dual SG/PB localization and modulates RNA translation or degradation. TTP and BRF1 are zinc-finger proteins involved in degradation of AU-rich element (ARE)–containing mRNAs (Lai et al., 1999; Stoecklin et al., 2002), and CPEB modulates cytoplasmic polyadenylation during development (Hake and Richter, 1994). TTP increases and stabilizes SG:PB docking, and at higher levels of expression forms fused granules where distinct PBs are contained within an SG. Overexpression of BRF1 has a similar effect, however, it induces the formation of fewer very large PBs within a given SG (Kedersha et al., 2005). In both cases there are two separate phases—PBs and SGs—however, one is embedded in the other. CPEB1 overexpression also increases SG:PB docking and forms hybrid granules. Hybrid granules induced by UBAP2L overexpression reflect mixing and increased connectivity of proteins and mRNAs (Sanders et al., 2020). Increased SG:PB docking and fusion is likely to reflect increased movement of mRNPs between SGs and PBs (Stoecklin and Kedersha, 2013), which is plausible given evidence of bidirectional mRNA movement between SGs and PBs (Moon et al., 2019).

Docking, fusion, or nucleation of hybrid granules upon protein overexpression suggests that these RBPs likely associate with both SGs and PBs when expressed at endogenous levels, even if only transiently. These transient interactions may allow for transfer of RNA or protein from one granule to another. Since each of these proteins mediating SG–PB interaction, including UBAP2L, modulates translation or RNA decay (Hake and Richter, 1994; Luo et al., 2020; Stebbins-Boaz et al., 1999), stress-responsive SG:PB docking and gene expression regulation may be connected. In fact, TTP's contribution to mRNA sorting is already documented. Normally, TTP facilitates the rapid degradation of ARE-mRNAs encoding for proteins that are needed on an acute basis, such as during the immune response, and require rapid degradation to maintain homeostasis. However, arsenite stress inhibits TTP-mediated degradation. Consequently, TTP delivers ARE-mRNAs to PBs during stress (Franks and Lykke-Andersen, 2007), which likely prevents them from being translated during stress when TTP cannot perform its normal function. This is a sequence-specific mechanism by which TTP regulates the fate of specific mRNAs during stress.

UBAP2L may also constitute an important part of gene expression regulation under stress; however, UBAP2L differs from TTP and other docking-mediating proteins in important ways. UBAP2L is a large, highly conserved, and abundant protein found in many cell types, whereas TTP is a small cell-type-specific protein. TTP, BRF1, and CPEB's alteration of SG–PB interaction have only been documented as a result of their overexpression (Kedersha et al., 2005; Wilczynska et al., 2005), while UBAP2L is clearly required for SG–PB interactions at endogenous levels (Fig. 1). This suggests UBAP2L's role in

docking may be physiologically relevant. UBAP2L is also a much more static protein, taking ~3 min to repopulate an SG after bleaching (Cirillo et al., 2020) compared to the 10–30 s required for TTP to recover (Ivanov et al., 2019; Kedersha et al., 2005). Rapid turnover of an SG protein suggests a role in shuttling or modifying mRNA, while slower turnover may indicate a role in stabilizing the granules or their contacts. We predict that UBAP2L helps stabilize and maintain transient interactions with PBs, which can last a few minutes (Kedersha et al., 2005). Like TTP, UBAP2L may also sort mRNAs on a sequence-specific basis; however, a specific class of mRNAs regulated by UBAP2L has not yet been identified. Since UBAP2L modulates regulators of global translation under optimal conditions (Luo et al., 2020), it may be that successful recovery from stress and resumption of translation requires UBAP2L-mediated sorting of these mRNAs. Alternatively, UBAP2L may modulate a different set of mRNAs under stress.

Though the function of SG–PB docking in human cells has yet to be identified, a functional significance of condensate interaction is not unprecedented. Interaction between PBs and P granules in *C. elegans* germ cells is important for transgenerational gene silencing. This is mediated by CGH-1, the homolog of DDX6, which contributes to proper separation and spatial arrangement of PBs and P granules, and to the organization of small RNA factors within these condensates (Du et al., 2023). In yeast, PB formation may be critical for SGs to form, although conflicting reports have been published (Buchan et al., 2008; Shah et al., 2013). Recent work proposes a model in which mRNAs may need to first transit through PBs to form SGs, also facilitated by DDX6 (Hondele et al., 2019). These examples demonstrate that biocondensate interaction may contribute to the formation of additional condensates, and that their proper formation and interactaction can have biological significance. Interestingly, DDX6 features prominently in the aforementioned examples and binds to UBAP2L. Thus, we propose that UBAP2L might work in concert with DDX6 to facilitate proper condensate interaction and function. Like DDX6, UBAP2L exhibits condition-specific recruitment to SGs or PBs.

UBAP2L also localizes to PBs. UBAP2L behaves similarly to DDX6 in this manner, suggesting dual localization is an important characteristic for modulation of condensate interaction. Our data confirm UBAP2L's recruitment to SGs under standard SG-inducing conditions (Huang et al., 2020; Sanders et al., 2020) (Fig. 1) and show UBAP2L is present in PBs in conditions yielding PBs but not SGs (Fig. 2). In G3BP1/2 KO cells treated with arsenite, the stoichiometry of the protein interaction network has been altered such that SGs are absent so that the unchanged "pull" of UBAP2L toward PBs predominates, resulting in UBAP2L condensing into PBs (Fig. 6). G3BP1/2 KO essentially unbalances the protein interaction network by removing the SG node (Sanders et al., 2020), thus tilting the system toward PBs. Other

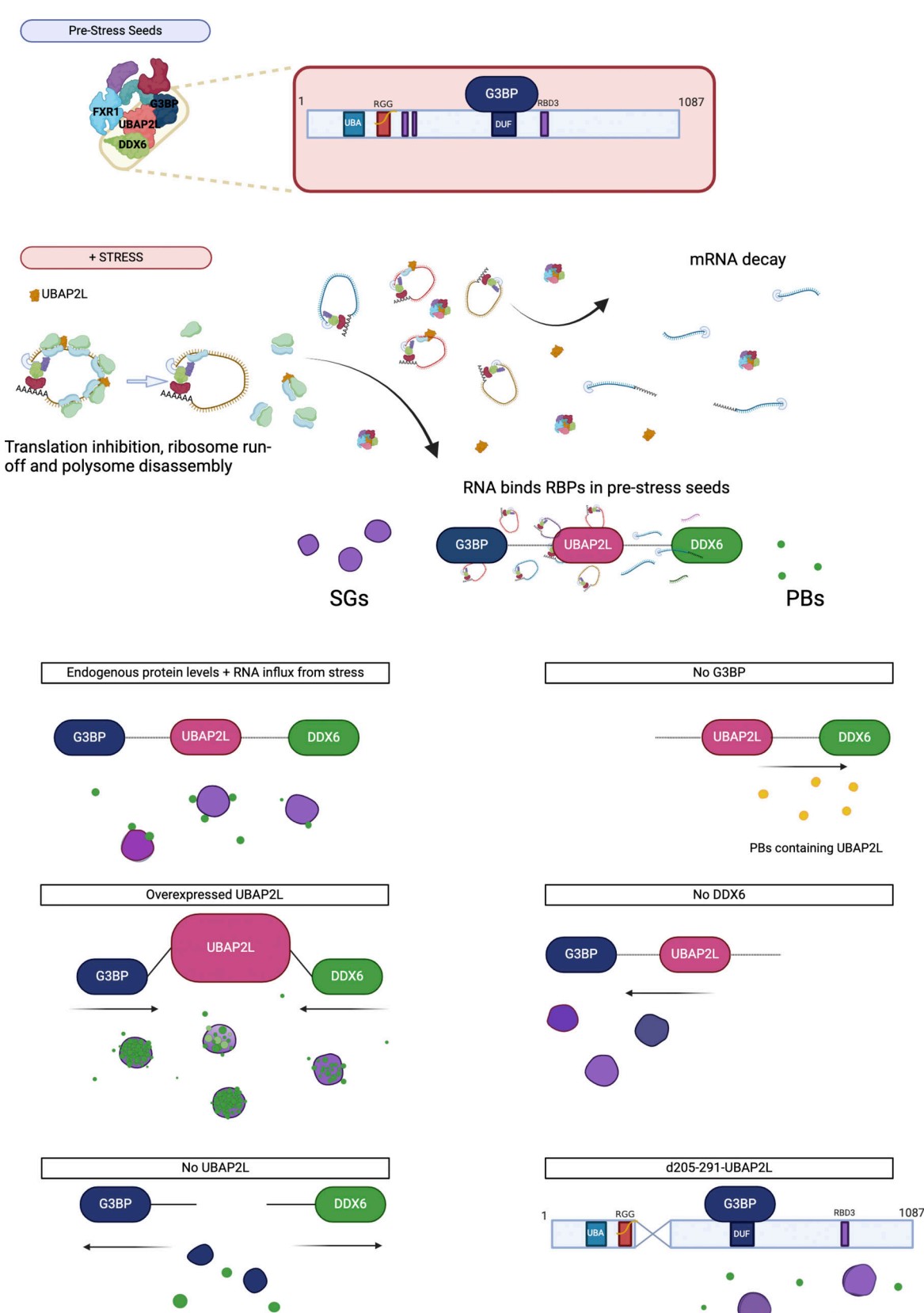

Figure 6. **Model of UBAP2L at the intersection of SGs and PBs.** UBAP2L and its SG- and PB-associated binding partners G3BP1, FXR1, and DDX6 are found in pre-stress protein complex seeds. UBAP2L also associates with polysomes. Stress inhibits translation initiation, resulting in ribosome run-off and freeing stalled pre-initiation complexes, which includes bound UBAP2L. Released mRNA is bound by additional RBPs in the pre-stress seed complexes, facilitating LLPS and giving rise to PBs and/or SGs. UBAP2L levels affect granule formation, interaction, and composition. In cells expressing endogenous levels of UBAP2L,

G3BP1, and DDX6, SGs with docking PBs form and UBAP2L localizes to SGs. Overexpression of UBAP2L induces hybrid granules containing canonical SG and PB components by essentially melding SGs and PBs together. In the absence of UBAP2L, fewer SGs and PBs form and SG:PB docking is reduced, as its connection between SGs and PBs is absent. Changes to protein and RNA concentrations also alter UBAP2L's localization. In cells lacking G3BP or DDX6, essential proteins for SG or PB formation, UBAP2L localizes to whichever type of condensate predominates. UBAP2L-mediated SG:PB docking requires amino acids 205–291, which contain two predicted RNA binding regions. Blue = G3BP; green = DDX6; magenta = UBAP2L; purple = mixture of G3BP (blue) and UBAP2L (red); yellow = mixture of UBAP2L (red) and DDX6 (green). As shown by purple and green, UBAP2L is in SGs when expressed at or beyond endogenous levels and is recruited to PBs in the absence of G3BP or conditions with less free mRNA. Components are not to scale. Protein ovals of the same size represent proteins expressed at their endogenous levels, but do not reflect relative protein stoichiometry. An increase in size of the protein oval (i.e., for UBAP2L) indicates expression above endogenous levels. Figure prepared using https://Biorender.com.

proteins can relocalize depending on the cellular conditions as well. For example, constitutive PBs—present in the absence of stress—contain YB1; during stress, YB1 relocalizes to SGs (Kedersha and Anderson, 2007).

SGs and PBs are heterogeneous entities, exhibiting stress and cell-type-specific composition (Advani and Ivanov, 2020; Aulas et al., 2017; Fujimura et al., 2012; Markmiller et al., 2018; Wang et al., 2018), which have been linked to differences in cell survival (Aulas et al., 2018; Fujimura et al., 2012). Our data showing UBAP2L localizes to PBs only under stress—and more so when SGs are absent—reveals possible PB subtypes dependent on the cellular conditions (Kedersha et al., 2005; Stoecklin and Kedersha, 2013). Localization of different proteins, such as UBAP2L, to PBs may reflect differences in the RNA composition of the granules as well, which could have functional consequences.

Based on our findings, we propose a model of how stress and UBAP2L levels uniquely modulate the formation of and interaction between SGs and PBs, as well as the subcellular distribution of UBAP2L (Fig. 6). Prior to stress, UBAP2L associates with many other SG and PB proteins (Markmiller et al., 2018; Marmor-Kollet et al., 2020) as well as with polysomes and monosomes (Luo et al., 2020). During stress, mRNAs released from polysomes interact with SG and PB proteins to form submicroscopic RNP complexes that include UBAP2L and subsequently condense into membraneless bodies through LLPS. SGs are driven by the influx of mRNAs released from translation but require G3BP to facilitate condensation of the RNA and proteins (Guillén-Boixet et al., 2020; Sanders et al., 2020; Yang et al., 2020). UBAP2L increases the RNA binding capacity of G3BP (Sanders et al., 2020), but is not absolutely required to drive SG formation, as UBAP2L KO cells still form some SGs. Similarly, UBAP2L increases the number of PBs formed but is not their primary driver. UBAP2L does appear critical for driving SG–PB interactions, likely via protein partners that bind to mRNAs. Given UBAP2L's long residence in SGs and its presence in pre-stress seeds, we propose that UBAP2L contributes to SG:PB docking under physiological conditions and is important for stabilizing interactions between SGs and PBs. Stabilizing these interactions may constitute an important aspect of gene expression regulation, allowing mRNAs to be exchanged or sorted between the granules.

Our model predicts that changes in the concentration of mRNAs or of specific proteins in the SG–PB network shift the equilibrium such that different bioconensates result (Fig. 6). In cells expressing proteins at endogenous levels, stress induces the accumulation of free mRNA and triggers the condensation of

both SGs and docking PBs. In cells overexpressing UBAP2L, its interactions with SG and PB nucleating proteins such as G3BP, FXR1, and DDX6 (Di Stefano et al., 2019) serve to pull PBs and SGs together—to the extent that PBs may merge with SGs. In contrast, UBAP2L depletion disrupts docking, reducing the connection between SGs and PBs (Fig. 6). Changing other protein levels also modulates the effect of endogenous proteins on RNA–protein condensation; KO of DDX6 forms hybrid granules characteristic of UBAP2L overexpression (Majerciak et al., 2023). Loss of DDX6 may imbalance the network equivalent to that of UBAP2L overexpression, effectively increasing the impact of UBAP2L on the system.

Likewise, changing relative concentrations of SG–PB network proteins not only alters condensate formation but also alters the subcellular localization of other proteins in the network, explaining UBAP2L's PB localization. Depleting the system of G3BP or DDX6 blocks the respective formation of SGs or PBs, and in both cases UBAP2L associates with whichever condensates are available, consistent with the intermediate position of UBAP2L in the SG–PB continuum. Importantly, UBAP2L-positive granules induced by arsenite in G3BP1/2 KO cells were recently described as "UBAP2L cores" and proposed to form upstream of G3BP in a linear model of SG formation (Cirillo et al., 2020). However, our results now show that such UBAP2L-positive foci in G3BP1/2 KO cells are PBs containing excess UBAP2L. Given that SG formation does not require the presence of PBs (Kedersha et al., 2005), we propose that UBAP2L does not form SG precursors prior to G3BP recruitment. Rather, UBAP2L promotes the condensation of SGs near PBs and facilitates the docking of PBs to SGs.

In summary, uncovering UBAP2L's role in PB biology reframes our understanding of stress-responsive mRNA biocondensates and their formation. Establishing UBAP2L as not only an SG-localized and nucleating protein, but rather a protein with dual roles in SGs and PBs, clarifies that there is not a single linear trajectory driving SG formation, but rather a dynamic network that modulates a continuum of phase-separated phenomena (Fig. 6). This highlights the need to view the network as a whole—rather than solely focusing on SGs or PBs individually. This is important as the interaction between SGs and PBs and UBAP2L's role in this phenomenon may constitute a key part of the stress response. SG and/or PB regulation of gene expression has long been a favored hypothesis, though it has been challenging to directly show the impact of distinct granules on translation regulation. The unifying role of UBAP2L in the context of SG–PB interaction and gene expression regulation may offer a path forward through a more holistic view.

## Materials and methods

### Cell lines

U2OS cell lines (Table S1) were maintained at ~5% $CO_2$ at 37°C in DMEM supplemented with 20 mM HEPES, 10% FBS, 100 U/ml penicillin, and 100 μg/ml streptomycin. SG KO U2OS cell lines (G3BP1/2 KO, UBAP2L KO, and FFF KO) were previously established through CRISPR-Cas9 technology (Kedersha et al., 2016; Sanders et al., 2020) (Table S1). Tetracycline-inducible (t/o) cell lines were established as follows (Table S1). Cells were seeded in 6-well plates and transfected with 2 μg DNA of t/o plasmids (Table S4) overnight. Following a brief recovery period in fresh media (~4–10 h), zeocin selection at 0.25 μg/ml was started. Zeocin-containing media was changed every 3–4 days for about 2 wk to complete drug selection. Once drug selected, cells were subcloned by limiting dilution and then screened by fluorescence microscopy and western blotting. For additional recommendations for generating stable cell lines expressing SG or PB markers, please see Kedersha et al. (2008). Though cell lines were grown from single clones, it is important to note that they still exhibit some cell-to-cell variability in their t/o expression. Additionally, each KO cell line and t/o cell line was established in the U2OS-WT Tet Repressor line. In any experiments using this background, the U2OS Tet Repressor line was used as the control. However, in experiments without tet-induced expression, U2OSWT cells were used. Unpublished data show no difference in the stress response between U2OS cells with and without the tet-repressor background.

### Stress treatment

Cells were stressed with sodium arsenite (100 or 250 μM, 15–90 min, depending on the experiment), sorbitol (0.1 M, 60 min), heat shock (45°C, 20 min), and NaCl (0.2 M, 30 min). All stress treatments were performed in preconditioned media. For heat shock, the 24-well plate containing cells was sealed with parafilm and floated in a water bath maintained at 45°C. For the recovery experiment, media was removed from wells after 1 h of arsenite treatment and replaced with standard DMEM media used for cell maintenance.

### Immunofluorescence staining

For immunostaining experiments, cells were seeded on #1.5 glass coverslips in 24-well plates at ~100,000 cells/well. Cells were stressed as indicated in the figure legends prior to fixation with 4% paraformaldehyde (Electron Microscopy Sciences) in PBS for 15 min at RT and permeabilized with pre-chilled absolute methanol for 5 min at RT. Cells were blocked in Blocking Solution (5% Normal Horse Serum in PBS with 0.02% sodium azide) for 1 h at RT. Primary antibodies (Table S2) were diluted with Blocking Solution and applied to cells for 1 h rocking at RT or overnight at 4°C. Secondary antibodies from Jackson Laboratories (Table S2) were diluted in Blocking Solution and incubated for 1 h rocking at RT. Samples were subsequently washed with PBS and mounted with polyvinol mounting media. Slides were viewed with a Nikon Eclipse E800 microscope (Nikon) with a 63× Plan Apo objective lens (NA 1.4) and illuminated with a mercury lamp and standard filters for Cy2 (FITC HQ 480/40; 535/50), Cy3 (Cy3 HQ 545/30; 610/75), and Cy5 (Cy5 HQ 620/60;

700/75) and imaged with a SPOT USB Pursuit Digital Camera (Diagnostic Instruments) using the manufacturer's software. Raw TIF files were compiled in Adobe Photoshop (CS5; Adobe Systems). FIJI (2.14.0) was used to change false color of images.

### SG, PB, and docking quantification

SG, PB, and docking quantifications shown in Fig. 1 were performed in FIJI. The following operations were performed with a macro coded in FIJI to facilitate reproducibility and consistent treatment of each sample. First, nuclei were detected by Auto Threshold using the Otsu algorithm on the Hoechst image. The nuclei were converted to a mask and used to exclude the nuclei from SG, PB, and docking counts. The SG image was processed by smoothing and performing a top hat filter with a radius of 7 to facilitate accurate capture of the SGs by Auto Thresholding with the MaxEntropy algorithm. Resulting particles that were 10 or more pixels in area and present outside the nuclei were counted as SGs. The PB image was prepared for thresholding to detect PBs by first performing a Gaussian subtraction. The PB image was duplicated. One image was filtered with a Gaussian Blur where sigma = 1, while the other was filtered with a Gaussian Blur where sigma = 2. The resulting sigma 2 blur image was subtracted from the sigma 1 blur image using the image calculator function in FIJI. The resulting image facilitated the accurate detection of PBs by MaxEntropy Auto Thresholding. Particles five or more pixels in the area and present outside of nuclei were counted as PBs. Masks of detected SGs and PBs were generated and used to identify docking between SGs and PBs. To do so, both masks were dilated once to slightly expand the area of each SG and PB. This allowed for edges of SGs and PBs in contact with each other to overlap. The two dilated images were used to create a mask of the overlapping areas, using the "AND" function in the region of interest (ROI) manager. The particle analyzer was run to count all overlapping areas between SGs and PBs to determine the number of docking PBs.

For PB quantification in G3BP1/2 KO cells, total cells versus total cells with PBs were manually counted per field ($n = 4$). Cells were considered PB-positive if three or more PBs (foci positive for HEDLS) were present. Graphing and statistical analyses were conducted in GraphPad Prism v9. For SG, PB, and docking quantification, an ANOVA with a Tukey's post-hoc test was performed to compare the means of cell types or conditions for each parameter measured. For PB quantification in G3BP1/2 KO cells, a one-way ANOVA with a Dunnett post-hoc test was used to compare each mean to that of the siGFP-treated cells.

### Quantification of protein recruitment to PBs and SGs

Images were taken on a Zeiss LSM 800 with Airyscan confocal system on a Zeiss Axio Observer Inverted Microscope at Brigham and Women's Hospital Confocal Microscopy Core with a 60× objective with oil. Z-stacks of 12 slices were taken spanning the depth of the granules (2.72 μm). For each protein, the settings were kept consistent for all samples analyzed within and across cell types and treatments (Fig. 2 A), as follows: Track 1 (cy3): 561 nm laser, 0.1% laser power, 730 V master gain, 47 μm pinhole; Track 2 (cy2): 488 nm laser, 0.1% laser power, 600 V master gain, 47 μm pinhole; Track 3 (cy5): 640 nm laser, 0.1%

laser power, 605 V master gain, 47 µm pinhole; and Track 4 (Hoechst): 405 nm laser, 0.2% laser power, 600 V master gain, 47 µm pinhole. All images were taken with 4× averaging of the mean intensity per line. Images were acquired at RT in the dark at using Zen 2.6 software. Each sample was imaged in three areas. Analysis was performed using FIJI. Samples were split into channels, converted to grayscale, and each stack was combined into one image comprised of the intensity of each slice summed together. This allowed us to capture the entirety of the signal from the foci of interest. The threshold tool was used to create ROIs for the foci of interest (PBs or SGs) and the cytoplasm. For quantification of UBAP2L presence in SGs and PBs (Fig. 2 B), HEDLS was used to create the PB ROI and eIF3b was used to create the SG ROI. These ROIs were then applied to the UBAP2L image to measure its intensity within PBs or SGs and the surrounding cytoplasm. Granules with mean intensity levels at least 3× higher than that of the cytoplasm were counted as UBAP2L-positive SGs or PBs. This analysis was performed on three images per condition, from which the mean and standard error of the mean were calculated. Means were compared to each other with a one-way ANOVA with Tukey's post-hoc test. Conditions with different letters indicate a statistically significant difference ($P < 0.05$).

To characterize the composition and identity of UBAP2L-positive foci in G3BP1/2 KO cells (Fig. 3 B), UBAP2L was used to define the ROI by thresholding. In cases where UBAP2L could not be included, due to limitations in antibodies from different species, HEDLS was used to define the foci area. These ROIs were then applied to each channel of the image to measure the mean intensity, area, and integrated density of each protein within the foci. To calculate the % recruitment to the foci of interest, the integrated density of the foci was divided by the integrated density of the cytoplasm (integrated density of the whole image—integrated density of the nuclei and background). This value was then divided by the % area occupied by the foci of interest to correct for differences in granule area from image to image. For each protein measured, at least three images from one independent experiment were used in the analysis. Some proteins are represented more times due to their detection on multiple samples (e.g., HEDLS and eIF3b). The mean signal in UBAP2L-positive foci was compared with a one-way ANOVA with Tukey's post-hoc test. Different letters indicate a statistically significant difference ($P < 0.05$). Proteins were also grouped by category (SG protein, PB protein, SG/PB dual localizing protein, and UBAP2L).

### Fluorescence in situ hybridization
Cells were seeded on #1.5 glass coverslips in 24-well plates at about 100,000 cells/well and allowed to incubate overnight. Cells were left untreated or treated with 250 µM sodium arsenite for 60 min prior to fixation with 4% PFA for 15 min at RT. Cells were then permeabilized with 0.1% Triton X-100 and 0.04% SDS in PBS for 15 min at RT. Cells were stored in 1 ml 70% EtOH at 4–8°C for 1–2 days. Cells were rehydrated with 2 × 10 min washes in 2× SSC (AM9763; Ambion) and prehybridized in prewarmed PerfectHyb Plus hybridization buffer (cat no. H7033; Sigma-Aldrich) at 42°C for 30 min. Cells were rinsed

twice in 2× SSC prior to hybridizing with 4 ng/ml 5′-cy3-Oligo-d(T)40 (IDT) probes in PerfectHyb Plus at 42°C for 1 h in a humid chamber. Hybridized coverslips were washed 3 × 10 min with prewarmed 2× SSC to 37°C. Cells were blocked with NHS-PBS blocking buffer and stained for HEDLS and eIF3b as described above in "Immunofluorescence staining".

### siRNA knockdowns
Approximately 100,000 cells were seeded per well in 6-well plates overnight. Cells were transfected with 50 nM siRNA for single gene knockdown or 25 nM each for depletion of two genes (Table S3) (as for simultaneous UBAP2 and UBAP2L knockdown) with 2.5 µl Lipofectamine 2000 (cat no. 11668019; Invitrogen). 24 h after transfection, cell culture medium was replaced with fresh medium. At 48 h, cells were transfected with a second dose of siRNA, as at t = 0. At ~72 h, cells were counted and reseeded in 12-well plates (~200,000 cells/well) and 24-well plates (~100,000 cells/well) for immunostaining and western blot analysis, respectively. Counting of cells at this stage is crucial. Due to the effect of lipofectamine and certain siRNAs on cell proliferation, cell densities were highly variable after siRNA treatments. High cell confluence can alter the ability to accurately quantify SGs and PBs. At ~96 h cells in 24-well plates were stress-treated (if desired) and harvested for immunofluorescence or western blotting.

### Western blotting
Cells were lysed via direct lysis using a 2X SDS lysis buffer (30% glycerol, 1 M Tris-Cl [pH 6.8], 6 mM EDTA, 10% SDS, 25 mM DTT, and 0.12 mg/ml Bromophenol Blue) and transferred to Eppendorf tubes. Samples were vortexed, sonicated 2 × 2 min, and boiled for 10 min. 10 µl sample was loaded on a precast polyacrylamide gradient gel (4–20% Mini-PROTEAN TGX Precast Protein Gel, #4561096; Bio-Rad) and run for 65 min at 140 V. Transfer was conducted with a Trans-Blot Turbo Transfer System (#1704150; Bio-Rad) using the high molecular weight setting onto a 0.2-µm nitrocellulose membrane (#1704270; Bio-Rad) for blotting. Membranes were blocked in ~5% non-fat dried milk in Tris-Buffered Saline with Tween-20 (TBST) at RT for 1 h or overnight at 4°C prior to incubation in primary antibody (Table S2) diluted in 5% NHS/PBS with 0.02% sodium azide at RT for 1 h or overnight at 4°C. Blots were rinsed with TBST 5 × 5–10 min and incubated in Peroxidase-Conjugated AffiniPure Donkey anti-Mouse or anti-Rabbit secondary antibodies (Table S2) at 1:5,000 in 5% NHS/TBST at RT for 1 h or overnight at 4°C. Blots were imaged using the ChemiDoc Imaging System (Bio-Rad). Quantification by band densiometry was performed in FIJI with UBAP2 or UBAP2L band intensity normalized to beta actin signal in the same lane, after subtracting background signal from each sample. Blotting and quantification were performed on four independent experiments ($n$ = 4). Changes in UBAP2 and UBAP2L abundance were calculated relative to the "no siRNA" treatment, which received the same amount of lipofectamine and optimem but no siRNA. The mean change in protein level was compared across treatment groups by an ANOVA with a Tukey post-hoc test.

## UBAP2L titration and PB analysis

t/o-GFP-UBAP2L cell lines were treated with dox to induce UBAP2L expression. For overexpression, cells were treated with 100 ng/ml dox ~24 h. To observe effects of different levels of UBAP2L expression, cells were treated with 10 ng/ml dox for 0, 5, 8, or 24 h. PB number and size were analyzed under these conditions as follows. PBs were selected using the threshold function in FIJI on the HEDLS channel, after filling the nuclei with black to avoid capturing nuclear signal. After adjusting the threshold to capture the PBs, PB size and number were obtained using the Analyze Particles function to report the size and mean intensity of particles 2–50 pixels in size. This threshold was necessary to exclude large inclusions of PB proteins in SGs that yielded SG-sized granules comprised of homogenous PB signals or PBs too close together to meaningfully distinguish. Analyses were performed on three to four images from one independent experiment. Quantification of HEDLS signal intensity in cells without UBAP2L (0 h dox) or overexpressing UBAP2L (24 h dox) was performed as follows. In each condition, SG ROIs were generated using the threshold tool on eIF3b. and applied to the PB image (HEDLS) to measure the mean intensity of each protein within SGs, with and without UBAP2L overexpression. An additional ROI was generated that excluded any background (area without cells), nuclei, SGs, and PBs. This ROI was used to measure the intensity of each protein in the cytoplasm. The mean intensity of HEDLS inside SGs relative to the mean intensity of HEDLS outside the granules was measured on three images for each treatment. To determine the % of PB area overlapping with SGs, the SG ROI was applied to the HEDLS image and filled in black to mask the SGs. PB measurements were taken with and without SGs occluded to calculate the % change in the total PB area visible.

## Transient transfection

Cells were seeded in 6-well plates and transfected with 2 μg plasmid DNA (Table S4) with 10 μl lipofectamine when 40–80% confluent. Cells were transfected overnight (~18 h) and allowed to recover in fresh culturing medium for ~4 h prior to re-seeding in 24-well plates on coverslips. On the next day, cells in 24-well plates were treated and harvested for immunofluorescence as described above.

All plasmids from David Sanders (UT Southwestern, Dallas, TX, USA) were generated using the FM5 lentiviral vector, though in this study they were transiently transfected. FM5 features the Ubiquitin C promoter. DNA fragments of interest were amplified by PCR and inserted into the vector using the In-Fusion HD cloning kit (Takara) (Sanders et al., 2020). Plasmids that have been used in prior publications are indicated in Table S4. Additional plasmids generously gifted to us by David Sanders were generated as a part of the same study (Sanders et al., 2020), but were not included in the publication.

## Expression of UBAP2L fragments in UBAP2L KO cells and quantification

Analysis was performed on individual cells expressing the GFP-UBAP2L fragment of interest (full-length UBAP2L, amino acids 91–1,087, 205–1,087, 291–1,087, or 495–1,087). For each cell analyzed, the number of SGs (defined as granules positive for UBAP2L and G3BP), hybrid granules (UBAP2L, G3BP, and DDX6 positive), PBs (distinct puncta positive for DDX6), and PBs docking to SGs were counted. $N$ = at least 5 cells per construct for one to two independent experiments. Cells from two biological replicates expressing full-length UBAP2L, amino acids 205–1,087, and 291–1,087 and cells from one biological replicate were counted for 91–1,087 and 495–1,087. 528–1,087 was not included in the analysis because UBAP2L remained in the nucleus and SGs did not form.

## Statistical analysis

As described above, means were compared to each other by ANOVA or $t$ test depending on the number of conditions being compared. All ANOVAs were one-way ANOVAs performed assuming Gaussian distribution and equal variance of the data; however, neither was formally tested. A post-hoc test was performed with each ANOVA to determine statistical differences among the conditions. All $t$ tests performed were two-tailed $t$ tests used to compare the means of two conditions. Throughout the manuscript, results are plotted as means with the standard error of the mean.

## Online supplemental material

Fig. S1 shows efficient UBAP2 and UBAP2L depletion by siRNA and provides additional consequences of depletion on SG and PB formation and UBAP2 localization. Fig. S2 further supports a role for UBAP2L in PB formation, showing this in G3BP 1/2 KO cells. Fig. S3 uses additional PB markers (EDC3 and DCP1A) to confirm the presence of UBAP2L in PBs under certain conditions. Fig. S4 examines the recruitment of numerous SG and PB proteins to UBAP2L-nucleated hybrid granules and shows their sequential formation over time. Fig. S5 uses UBAP2 and UBAP2L protein alignment to identify highly conserved regions. Effects of deletion or expression of these regions on SG and PB interaction are detailed. Table S1 shows cell lines used in this study and their sources. Table S2 shows information about antibodies used in this study. Table S3 shows details of siRNAs used in this study. Table S4 shows plasmids used to generate t/o cell lines or for transient transfection in this study.

## Data availability

The data are available from the corresponding author upon reasonable request.

## Acknowledgments

We thank Tiffany Ye (UMass Boston, Boston, MA, USA) for her assistance with image quantification, Ashley Tai (University of Rhode Island, South Kingstown, RI, USA) for her progress toward understanding the role of different domains of UBAP2L, and Dr. David Sanders (UT Southwestern, Dallas, TX, USA) and Dr. Cliff Brangwynne (Princeton University, Princeton, NJ, USA) for sharing their UBAP2L constructs with us. We wish to thank the Brigham and Women's Hospital Confocal Microscopy Core (Boston, MA, USA) for the use of the Zeiss LSM 800 with Airyscan. We thank Beth Beighlie, Research Computing Specialist at

the Department of Information Technology at Harvard Medical School (Boston, MA, USA), for assistance in developing workflows using best practices for figure generation from microscopy images. We thank members of the Image Analysis Collaboratory, in particular Drs. Maria Theiss and Ranit Karmakar, at Harvard Medical School (Boston, MA, USA) for invaluable consultation on SG and PB quantification in FIJI. We thank members of the Ivanov/Anderson lab for critical feedback on the project. Finally, we are grateful to the three anonymous reviewers for their time and expertise; their comments have greatly improved the manuscript.

Research reported in this publication was supported by the National Institutes of General Medical Sciences of the National Institutes of Health F32GM142262 and K99GM148723 to C.L. Riggs, R01 GM126150 and R01 GM146997 to P. Ivanov, and R35GM12901 to P. Anderson.

Author contributions: C.L. Riggs: Conceptualization, Data curation, Formal analysis, Funding acquisition, Investigation, Methodology, Project administration, Software, Supervision, Validation, Visualization, Writing - original draft, Writing - review & editing, N. Kedersha: Conceptualization, Investigation, Supervision, Writing - review & editing, M. Amarsanaa: Investigation, S.N. Zubair: Data curation, Formal analysis, Investigation, Software, Visualization, P. Ivanov: Conceptualization, Formal analysis, Funding acquisition, Project administration, Resources, Supervision, Validation, Writing - original draft, Writing - review & editing, P. Anderson: Conceptualization, Funding acquisition, Resources, Supervision, Writing - original draft, Writing - review & editing.

Disclosures: P. Anderson reported personal fees from Simcere USA outside the submitted work. No other disclosures were reported.

Submitted: 28 July 2023

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

**Supplemental material**

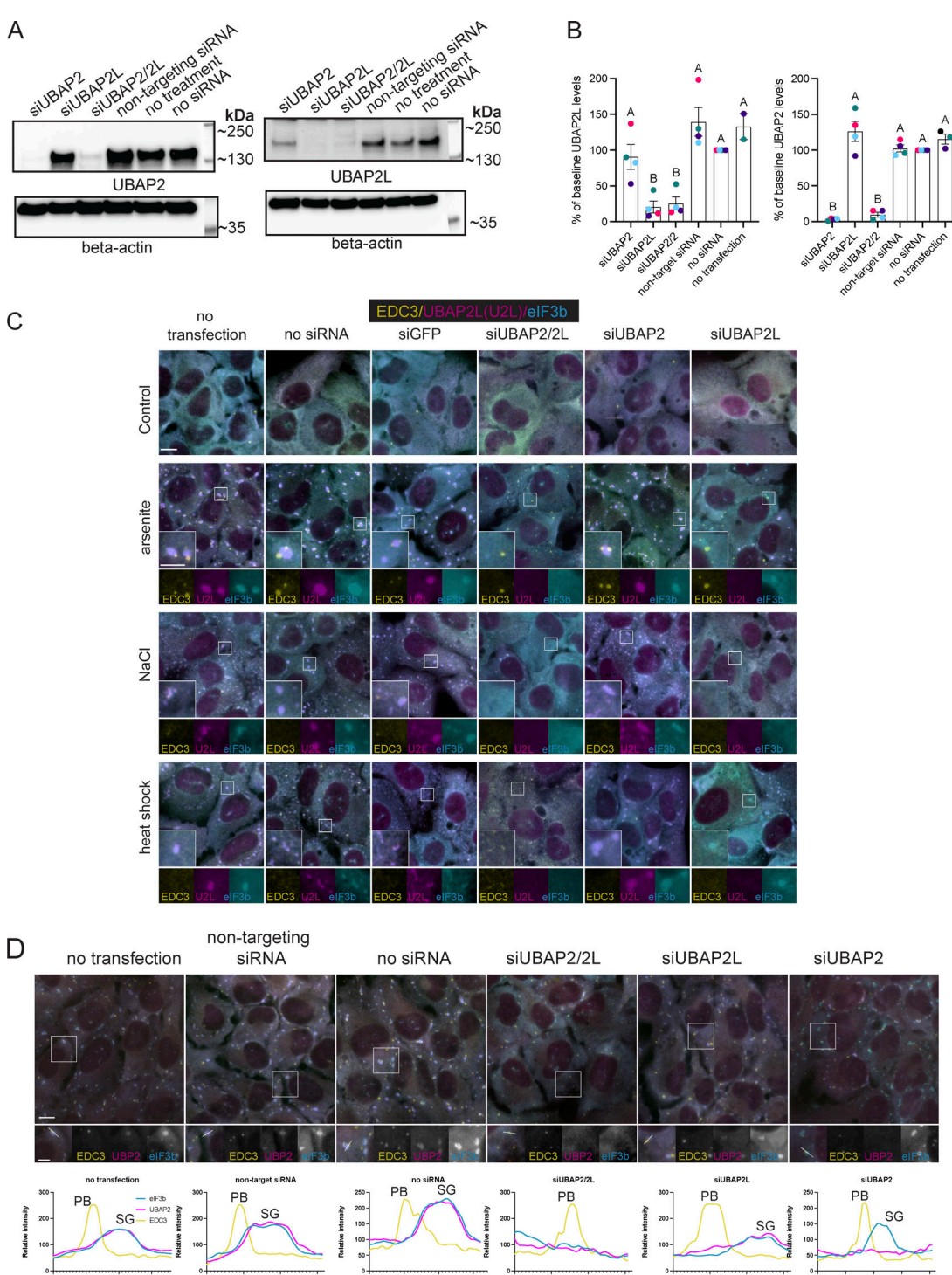

Figure S1. **Efficient depletion of UBAP2 and UBAP2L by siRNA and consequences for SG formation and protein localization. (A)** Representative western blot of siRNA-treated samples (no siRNA, non-targeting siRNA, siUBAP2, siUBAP2L, siUBAP2/2L) probed for β-actin and UBAP2L or UBAP2 specific antibodies. **(B)** Quantification of siRNA experiment ($n$ = 4 independent experiments). Different colored symbols represent results of independent experiments. Data are plotted as the mean percent change in UBAP2 or UBAP2L abundance for each siRNA treatment. Error bars show the standard error of the mean. A one-way ANOVA with Tukey's post-hoc test was used to compare the mean protein level across siRNA treatments. siUBAP2 and siUBAP2L effectively knock down target proteins. siUBAP2L and siUBAP2/2L significantly reduce UBAP2L levels from that of cells treated with non-targeting siRNA. siUBAP2 and siUBAP2/2L significantly reduce UBAP2 levels from that of cells treated with non-targeting siRNA. **(C)** Representative images of U2OS-WT cells with siRNA depletion of UBAP2, UBAP2L, or UBAP2/UBAP2L followed by osmotic stress (0.2 M NaCl, 30 min) or heat shock (45°C, 20 min). Yellow = EDC3 (PB marker); magenta = UBAP2L; cyan = eIF3b (SG marker). **(D)** Representative images of U2OS-WT cells treated with siRNA (as in Fig. 1 C), treated for 60 min with 250 μM sodium arsenite, and immunostained for UBAP2 (UBP2), EDC3, and eIF3b. Intensity profile plots shown below each image correspond to the white arrow drawn through a PB and SG (if present) on the image. Immunostaining and plot colors match: magenta = UBAP2; yellow = EDC3 (PB marker), cyan = eIF3b (SG marker). Intensity profiles show UBAP2 in SGs regardless of UBAP2L depletion. Source data are available for this figure: SourceData FS1.

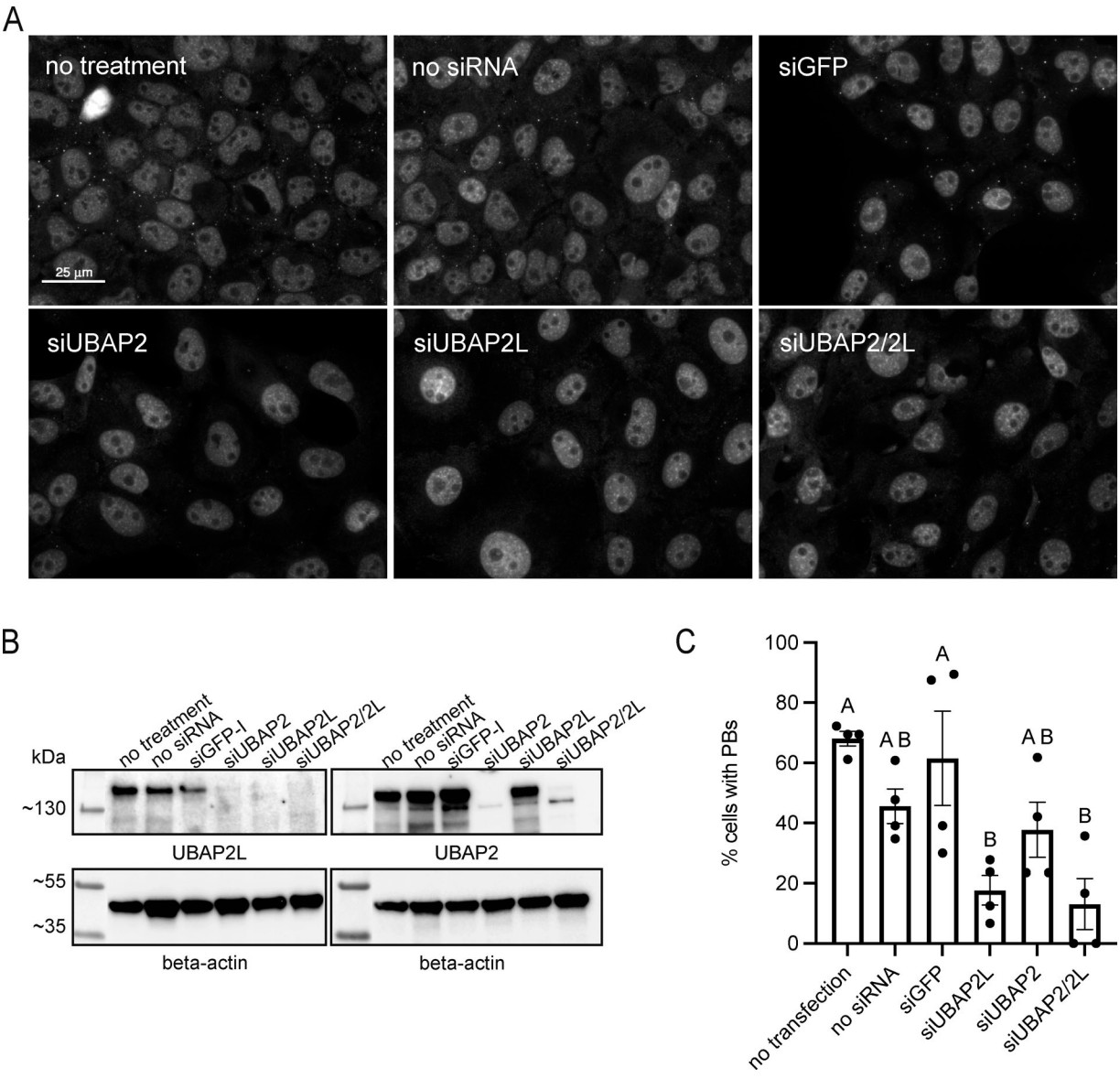

Figure S2. **UBAP2L contributes to PB formation in G3BP1/2 KO cells. (A)** UBAP2L contributes to PB formation in G3BP1/2 KO cells. Representative images show G3BP1/2 KO cells treated with siRNA targeting UBAP2L, UBAP2, UBAP2/2L, or GFP, treated with arsenite (100 µM, 60 min) to induce PBs, and immunostained for HEDLS to detect PBs. No treatment = no siRNA or transfection reagents. **(B)** Representative western blot shows efficient depletion of UBAP2 and UBAP2L in G3BP1/2 KO cells. **(C)** Quantification of percentage of cells with PBs. For each treatment, four images from one independent experiment were analyzed. Cells with at least three PBs were counted as PB-positive cells. Data were analyzed with a one-way ANOVA with a Tukey's post-hoc test to compare mean % PB-positive cells among treatments. Treatments with no letters in common are significantly different (P < 0.05).

Figure S3. **Additional PB markers confirm UBAP2L localization to PBs under certain conditions. (A and B)** Representative images of U2OS-WT and G3BP1/2 KO cells treated with arsenite (250 µM, 60 min) or sorbitol (0.1 M, 60 min), and immunostained for UBAP2L (magenta), SG marker eIF3b (cyan), and two PB markers (yellow), EDC3 (A) and DCP1a (B). UBAP2L is recruited to EDC3 and DCP1a-positive foci (PB markers) in conditions where SGs are absent, as observed with HEDLS as the PB marker (Fig. 2 A).

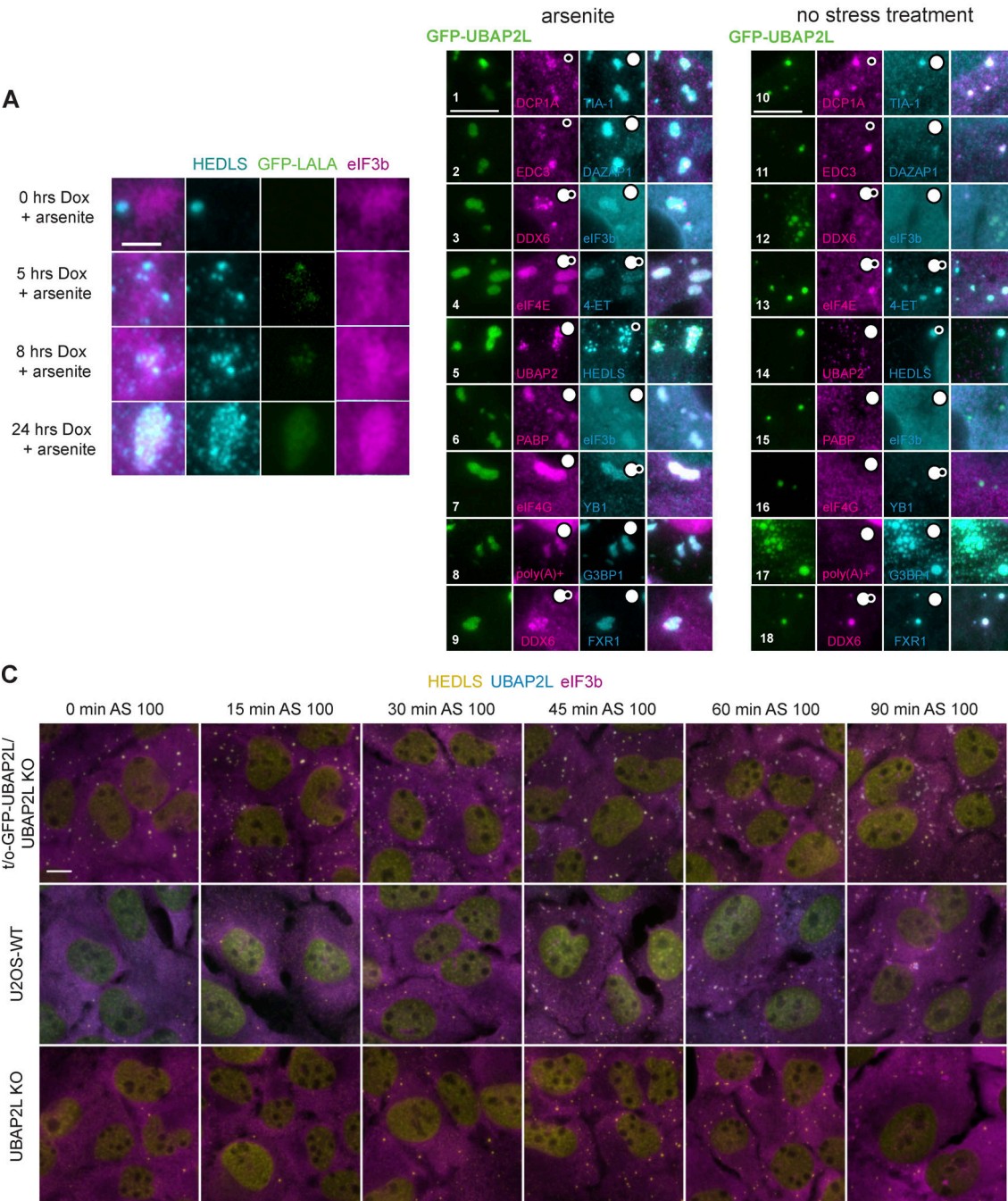

Figure S4. **Composition and formation of hybrid granules nucleated by UBAP2L overexpression. (A)** UBAP2L-LALA can regulate SG–PB association under stress, like UBAP2L-WT. Tet-inducible GFP-UBAP2L-LALA/UBAP2L KO (GFP-LALA) U2OS cells were treated with 100 ng/ml dox for 0, 5, 8, or 24 h to induce varying levels of UBAP2L-LALA prior to treatment with arsenite (250 µM, 60 min). LALA is a mutant version of UBAP2L unable to bind G3BP1 (Baumgartner et al., 2013; Youn et al., 2018). Following dox treatment, cells for immunostaining were treated with sodium arsenite (250 µM, 60 min). Magenta = eIF3b (SG marker); cyan = HEDLS (PB marker); green = GFP-UBAP2L. Scale bar = 5 µm. Representative images show formation of hybrid granules when UBAP2L-LALA is overexpressed. See Fig. 4 C for corresponding western blot. **(B)** UBAP2L overexpression recruits numerous SG and PB components to UBAP2L-positive granules. Tet-on-GFP-UBAP2L/UBAP2L KO cells were treated with dox (100 ng/ml, 24 h) to induce UBAP2L overexpression. Granule composition in arsenite-treated (250 µM, 60 min) and untreated cells overexpressing UBAP2L was analyzed by immunofluorescence and in situ hybridization to detect known SG and PB components. Green = GFP-UBAP2L. Magenta and cyan: see annotations on figure. Scale bar = 5 µm. **(C)** Arsenite time course showing timing and relative subcellular localization of PBs, SGs, and hybrid granules in U2OS-WT cells, UBAP2L KO cells, and t/o-GFP-UBAP2L/UBAP2L KO cells overexpressing UBAP2L. Cyan = eIF3b (SG marker); magenta = HEDLS (PB marker); yellow = GFP-UBAP2L or endogenous UBAP2L (U2OS-WT). Scale bar = 10 µm.

**A**

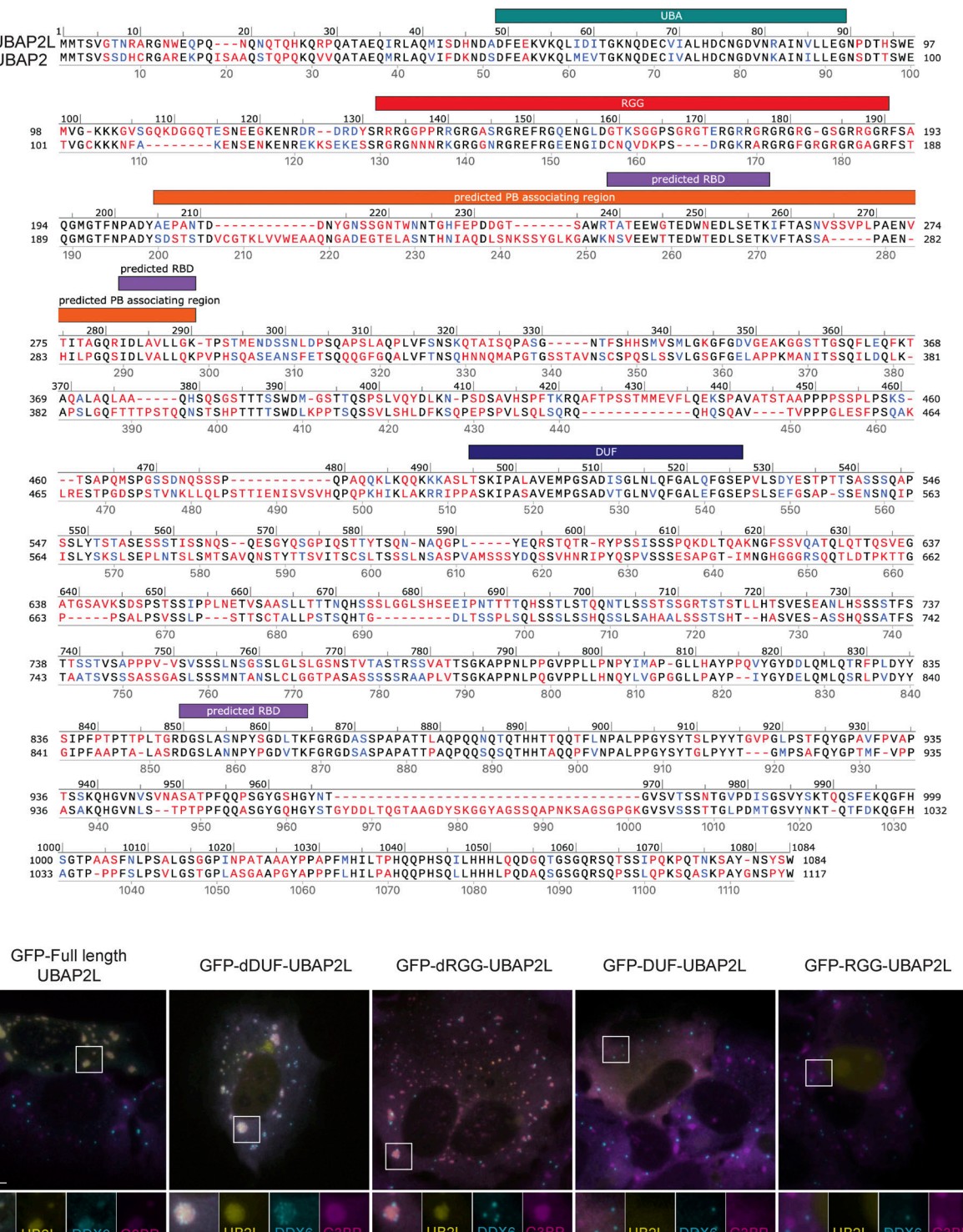

**B**

GFP-Full length UBAP2L GFP-dDUF-UBAP2L GFP-dRGG-UBAP2L GFP-DUF-UBAP2L GFP-RGG-UBAP2L

**Figure S5. UBAP2L and UBAP2 sequence alignment and contribution of conserved regions to hybrid granule formation. (A)** NCBI UBAP2L (UBAP2L isoform a [*Homo Sapiens*] NCBI Ref Seq: NP_055662.3) and UBAP2 (UBAP2 isoform 1 [*H. sapiens*] NCBI Ref Seq: NP_001356991.2) sequences were aligned in SnapGene using the Smith Waterman local alignment. Sequences are annotated with known and predicted region characterizations. Amino acid colors in alignment signify the following: same amino acid = black; similar amino acid = blue; very different or missing amino acid = red. Note the high conservation between UBAP2L and UBAP2 in UBA, RGG, and DUF regions and lower conservation in the predicted PB associating region (indicated in orange), based on Fig. 5. **(B)** Contribution of UBAP2L RGG and DUF domains to UBAP2L localization, SG:PB docking, and to hybrid granule formation. UBAP2L KO cells transiently transfected with GFP-UBAP2L-RGG, GFP-UBAP2L-DUF, GFP-UBAP2L-dRGG, GFP-UBAP2L-dDUF, or full-length UBAP2L were treated with arsenite (250 μM, 60 min). Immunostaining was performed to detect DDX6 (cyan) and G3BP1 (magenta). Yellow = GFP-labeled UBAP2L fragment. Scale bar = 10 μm in the main image and 5 μm in the inset.

Provided online are four tables. Table S1 shows cell lines used in this study and their sources. Table S2 shows information for antibodies used in this study. Table S3 shows details of siRNAs used in this study. Table S4 shows plasmids used to generate t/o cell lines or for transient transfection in this study.

