## [Peer Review File · The Journal of Cell Biology]

UBAP2L contributes to formation of P-bodies and modulates their association with Stress Granules

Claire Riggs, Nancy Kedersha, Misheel Amarsanaa, Safiyah Zubair, Pavel Ivanov, and Paul Anderson

Corresponding Author(s): Paul Anderson, Brigham and Woman's Hospital

Review Timeline:

Submission Date:	2023-07-28
Editorial Decision:	2023-09-15
Revision Received:	2024-03-06
Editorial Decision:	2024-04-01
Revision Received:	2024-05-29

Monitoring Editor: Karla Neugebauer

Scientific Editor: Dan Simon

Transaction Report:

DOI: <https://doi.org/10.1083/jcb.202307146>

September 15, 2023

Re: JCB manuscript #202307146

Prof. Paul Anderson
Brigham and Woman's Hospital
75 Francis St
Boston, MA 02115

Dear Prof. Anderson,

Thank you for submitting your manuscript entitled "UBAP2L: at the Intersection of Stress Granules and P-Bodies." Your manuscript has been assessed by expert reviewers, whose comments are appended below. Although the reviewers express potential interest in this work, significant concerns unfortunately preclude publication of the current version of the manuscript in JCB.

You will see that Reviewer #1 requests new experiments to clarify the physiological function of UBAP2L-containing PBs and the merged SG:PB granules as well as a few additional assays to strengthen the current data. Reviewers 2&3 request quantifications for many of the experiments and Reviewer #2 notes several relevant papers that are not cited. We agree with the reviewer comments and these would need to be thoroughly addressed in a revised manuscript. Confirming some of the findings in another cell type would enhance the paper but we do not think this is essential for revision.

Please let us know if you are able to address the major issues outlined above and wish to submit a revised manuscript to JCB. Note that a substantial amount of additional experimental data likely would be needed to satisfactorily address the concerns of the reviewers. The typical timeframe for revisions is three to four months. While most universities and institutes have reopened labs and allowed researchers to begin working at nearly pre-pandemic levels, we at JCB realize that the lingering effects of the COVID-19 pandemic may still be impacting some aspects of your work, including the acquisition of equipment and reagents. Therefore, if you anticipate any difficulties in meeting this aforementioned revision time limit, please contact us and we can work with you to find an appropriate time frame for resubmission. Please note that papers are generally considered through only one revision cycle, so any revised manuscript will likely be either accepted or rejected.

If you choose to revise and resubmit your manuscript, please also attend to the following editorial points. Please direct any editorial questions to the journal office.

GENERAL GUIDELINES:

Text limits: Character count is < 40,000, not including spaces. Count includes title page, abstract, introduction, results, discussion, and acknowledgments. Count does not include materials and methods, figure legends, references, tables, or supplemental legends.

Figures: Your manuscript may have up to 10 main text figures. To avoid delays in production, figures must be prepared according to the policies outlined in our Instructions to Authors, under Data Presentation, <https://jcb.rupress.org/site/misc/ifora.xhtml>. All figures in accepted manuscripts will be screened prior to publication.

Title: The title should be less than 100 characters including spaces. Make the title concise but accessible to a general readership. While your current title may be appreciated by specialists, we do not feel that it clearly conveys the advance and will be accessible to a broader cell biology audience.

Supplemental information: There are strict limits on the allowable amount of supplemental data. Your manuscript may have up to 5 supplemental figures. Up to 10 supplemental videos or flash animations are allowed. A summary of all supplemental material should appear at the end of the Materials and methods section.

Please note that JCB now requires authors to submit Source Data used to generate figures containing gels and Western blots with all revised manuscripts. This Source Data consists of fully uncropped and unprocessed images for each gel/blot displayed in the main and supplemental figures. Since your paper includes cropped gel and/or blot images, please be sure to provide one Source Data file for each figure that contains gels and/or blots along with your revised manuscript files. File names for Source Data figures should be alphanumeric without any spaces or special characters (i.e., SourceDataF#, where F# refers to the associated main figure number or SourceDataFS# for those associated with Supplementary figures). The lanes of the gels/blots

should be labeled as they are in the associated figure, the place where cropping was applied should be marked (with a box), and molecular weight/size standards should be labeled wherever possible. Source Data files will be made available to reviewers during evaluation of revised manuscripts and, if your paper is eventually published in JCB, the files will be directly linked to specific figures in the published article.

If you choose to resubmit, please include a cover letter addressing the reviewers' comments point by point. Please also highlight all changes in the text of the manuscript.

Regardless of how you choose to proceed, we hope that the comments below will prove constructive as your work progresses. We would be happy to discuss them further once you've had a chance to consider the points raised. You can contact the journal office with any questions, cellbio@rockefeller.edu or call (212) 327-8588.

Thank you for thinking of JCB as an appropriate place to publish your work.

Sincerely,

Karla Neugebauer, PhD
Monitoring Editor
Journal of Cell Biology

Dan Simon, PhD
Scientific Editor
Journal of Cell Biology

Reviewer #1 (Comments to the Authors (Required)):

In this study, Riggs and his/her co-authors characterize the novel roles of UBAP2L, an RNA-binding protein, in PB biogenesis and SG/PB interaction, besides its known role in SG assembly. They also decipher the protein architecture in PB association. The IF data are beautiful and convincing. The findings are novel and interesting, providing new insights to PB biology and SG/PB association, as well as to RNA metabolism in cells under friendly or adverse conditions. While there are also some concerns worthy of discussion.

1. Fig.1. UBAP2 also involved in SGs. In Fig.1C, it may be required to examine whether UBAP2 is also associated with PB using a UBAP2 antibody.
2. Fig. 1 aims to show that UBAP2L contributes to the formation of SGs and PBs and their interaction. In Fig. 1D, as UBAP2L is deleted, of course no UBAP2L-positive IF signal was observed, and of course no UBAP2L-SG and no coalescence of UBAP2L-SG and PB marker was observed. It might be more appropriate to use another SG marker for detecting SG/PB association following UBAP2L deletion, to address that UBAP2L contributes to PB formation. In this sense, Fig.2 is well-designed, with G3BP being KO and using UBAP2L as a marker.
3. In Fig. 4, when conclude that G3BP is not required for UBAP2L to form PBs, Fig.3 may also should be cited, as this figure at first use G3BP-KO cells.
4. Like SGs, PBs are also dynamic. It is still unknown whether UBAP2L is present in PBs and contributes to SG/PB interaction when stress is removed or delayed. And what is the contribution of UBAP2L under these conditions?
5. The significance of the presence of UBAP2L in PBs and SG/PB interaction seems still not clearly characterized, despite of some discussion. Dose it definitely contribute to SG/PB component interchange? Especially RNA for decay in PBs? Component chasing or interchange array in live cells may be a choice to answer this question.

Minor:

1. Table 2, please add the working dilution for WB. Besides, there is a repetition of the information for UBAP2L antibody.

Reviewer #2 (Comments to the Authors (Required)):

Here the authors suggest that the stress granule component UBAP2L has a role in processing bodies formation. While the manuscript is potentially interesting (more adequate for a report and not a full article), many of the experiments are not quantified, which makes the conclusion an overinterpretation. An additional problem is that the literature is cited incorrectly or not cited at all.

Detailed comments

Two major weaknesses of this manuscript are:

1) the authors only look at one cancer cell line (U2OS)

2) there are no quantifications for any of the major statements that follow figure 1. For example, what is the percentage of colocalization between UBAP2L granules and the other components shown in figure 2 in control and G3BP1/2 KO cells? Same is true for most following figures.

There is a lack of citations of literature in model organisms and below I cite a few but not all examples. While it is true that stress induces Processing Bodies formation, PBs in many organisms are always there, independently of any (known) stress (eg *S. cerevisiae*, plants, *Drosophila*, *C. elegans*). Most of the literature cited in the article only refers to cancer cell lines.

Literature in yeast suggests that PBs can be the seeds of SGs, which is very relevant in the context of the data presented here (Hondele et al., *Nature*, 2019).

In line 71-72, the authors say that RNAs in PBs lack detectable polyadenylation and cite Hubstenberger et al., 2017. However, in this article it is stated "Thus, poly(A) tails of P-body-enriched mRNAs are not shorter or longer on average, but are more heterogeneous. Altogether, P-body-targeted mRNAs are characterized by variable poly(A) tail lengths and poor translation. »

Line 130-131, page 5: the authors say that it is unclear if the UBA domain of UBAP2L contributes to SG formation and cite Huang, 2020 and Markmiller, 2018. To my knowledge, Marmiller has no data on the domains of UBAP2L and SG formation while both Huang et al., 2020 and Youn et al, 2018 show that the UBA domain is not required for SG formation.

Other statements have no reference et al, eg line 60-61, line 132-133

Line 216, page 8: The laboratory of Roy Parker has shown that not only poly(A) mRNAs are in SGs

Line 225, page 8: polysome disassembly is also required for SG formation

Line 246: did Huang et al measure the affinity of UBAP2L and Fragile X proteins?

Minor comments:

Lines 157-158: the authors state: "proximity labelling studies only identified the involvement....". Proximity labelling studies do not identify involvement, they only identify proteins (or other molecules) proximal to the bait.

Lines 192-193: A review is cited to support the statement

Line 265: here also the references seem a bit random

Reviewer #3 (Comments to the Authors (Required)):

The authors show that UBAP2L and its homolog UBAP2 both localise to SG in oxidative stress. However, KD of UBAP2 does not seem to modulate SG formation, so the authors focused on UBAP2L. KO of UBAP2L abolishes SG formation and surprisingly also reduces PB formation and PB-SG docking; expression of GFP-UBAP2L restores these defects. In cells lacking SG (WT with mild osmotic stress, or G3BP1/2 KO with arsenite) UBAP2L also localises to PB. As determined by IF, the foci formed in these SG-lacking cells contain known PB marker proteins (including proteins known to localise to both PB and SG), but lack SG marker proteins and polyA+ mRNAs. Overexpression of UBAP2L induces formation of granules that contain both PB and SG marker proteins in stress, but mostly PB components without stress. The authors further identify region 205-291, containing two RNA binding domains, as essential for PB-SG docking and merging.

The paper is very well written, and figures clearly labeled, with the introduction giving all background required even for non-experts, and the discussion putting the results nicely into context of the published literature.

Overall, the only major comment I have is that appropriate quantification of protein enrichment would greatly help the reader evaluate the original data, especially since in most cases only few cells are shown.

Figure S1: siUBAP2L seems to be reduced by siUBAP2. Please quantify and, if true, please mention in text or figure legend of the main figure

Figure 1C: the blue eIF3b signal as stress granule marker is hard to see in the merged images, the UBAP2L (red) signal is very strong. It would be good to have the split channel images, at least as a supplementary figure, to be able to judge the changes in the eIF3b signal. Please add a quantification (as in Figure 1E).

Figure 2A: as the authors write at the end of the paragraph, recruitment of UBAP2L to PB seems to be not as efficient in sorbitol;

some kind of quantification, e.g.' intensity measurements along a line through the MLO' for all three marker proteins would be very helpful.

Figure 3A: maybe color code the legend to help the reader (e.g. write "PB components" in green). The 'shared' component YB1 seems not to be recruited, also eIF4E seems to be less enriched than others; therefore I would suggest to find a way to quantify this data.

Figure 4A / 5A, 24h time point: While the SG marker proteins (UBAP2L, eIF4E, eIF4G, PABP) seems to form one large granule, the PB marker proteins (HEDLS, DDX6, EDC3 etc) seems to form foci within this granule. Could the authors please detail why they consider this 'uniform distribution' / or 'merged'? I would suggest to rephrase because at least to me, the term 'uniform' implies a fully mixed granule.

Figure 5A/5B: please quantify, and if possible color-code again PB and SG and shared marker proteins to make it easier for the reader.

Dear Prof. Anderson,

Thank you for submitting your manuscript entitled "UBAP2L: at the Intersection of Stress Granules and P-Bodies." Your manuscript has been assessed by expert reviewers, whose comments are appended below. Although the reviewers express potential interest in this work, significant concerns unfortunately preclude publication of the current version of the manuscript in JCB.

You will see that Reviewer #1 requests new experiments to clarify the physiological function of UBAP2L-containing PBs and the merged SG:PB granules as well as a few additional assays to strengthen the current data. Reviewers 2&3 request quantifications for many of the experiments and Reviewer #2 notes several relevant papers that are not cited. We agree with the reviewer comments and these would need to be thoroughly addressed in a revised manuscript. Confirming some of the findings in another cell type would enhance the paper but we do not think this is essential for revision.

Please let us know if you are able to address the major issues outlined above and wish to submit a revised manuscript to JCB. Note that a substantial amount of additional experimental data likely would be needed to satisfactorily address the concerns of the reviewers. The typical timeframe for revisions is three to four months. While most universities and institutes have reopened labs and allowed researchers to begin working at nearly pre-pandemic levels, we at JCB realize that the lingering effects of the COVID-19 pandemic may still be impacting some aspects of your work, including the acquisition of equipment and reagents. Therefore, if you anticipate any difficulties in meeting this aforementioned revision time limit, please contact us and we can work with you to find an appropriate time frame for resubmission. Please note that papers are generally considered through only one revision cycle, so any revised manuscript will likely be either accepted or rejected.

If you choose to revise and resubmit your manuscript, please also attend to the following editorial points. Please direct any editorial questions to the journal office.

GENERAL GUIDELINES:

Text limits: Character count is < 40,000, not including spaces. Count includes title page, abstract, introduction, results, discussion, and acknowledgments. Count does not include materials and methods, figure legends, references, tables, or supplemental legends.

Figures: Your manuscript may have up to 10 main text figures. To avoid delays in production, figures must be prepared according to the policies outlined in our Instructions to Authors, under Data Presentation, <https://jcb.rupress.org/site/misc/ifora.xhtml>. All figures in accepted manuscripts will be screened prior to publication.

Title: The title should be less than 100 characters including spaces. Make the title concise but accessible to a general readership. While your current title may be appreciated by specialists, we do not feel that it clearly conveys the advance and will be accessible to a broader cell biology audience.

*****IMPORTANT:** It is JCB policy that if requested, original data images must be made available. Failure to provide original images upon request will result in unavoidable delays in publication. Please ensure that you have access to all original microscopy and blot data images before submitting your revision. *******

Supplemental information: There are strict limits on the allowable amount of supplemental data. Your manuscript may have up to 5 supplemental figures. Up to 10 supplemental videos or flash animations

are allowed. A summary of all supplemental material should appear at the end of the Materials and methods section.

Please note that JCB now requires authors to submit Source Data used to generate figures containing gels and Western blots with all revised manuscripts. This Source Data consists of fully uncropped and unprocessed images for each gel/blot displayed in the main and supplemental figures. Since your paper includes cropped gel and/or blot images, please be sure to provide one Source Data file for each figure that contains gels and/or blots along with your revised manuscript files. File names for Source Data figures should be alphanumeric without any spaces or special characters (i.e., SourceDataF#, where F# refers to the associated main figure number or SourceDataFS# for those associated with Supplementary figures). The lanes of the gels/blots should be labeled as they are in the associated figure, the place where cropping was applied should be marked (with a box), and molecular weight/size standards should be labeled wherever possible. Source Data files will be made available to reviewers during evaluation of revised manuscripts and, if your paper is eventually published in JCB, the files will be directly linked to specific figures in the published article.

If you choose to resubmit, please include a cover letter addressing the reviewers' comments point by point. Please also highlight all changes in the text of the manuscript.

Regardless of how you choose to proceed, we hope that the comments below will prove constructive as your work progresses. We would be happy to discuss them further once you've had a chance to consider the points raised. You can contact the journal office with any questions, cellbio@rockefeller.edu or call (212) 327-8588.

Thank you for thinking of JCB as an appropriate place to publish your work.

Sincerely,

Karla Neugebauer, PhD
Monitoring Editor
Journal of Cell Biology

Dan Simon, PhD
Scientific Editor
Journal of Cell Biology

Dear Dr. Neugebauer and Dr. Simon,

We thank you for your careful consideration and constructive consideration of our work on UBAP2L's role in SG and PB formation and interactions. We have performed and added additional experiments and analyses, based on the reviewer's recommendations, significantly improving the manuscript.

These improvements include quantification of major findings (Fig 1-5), additional experiments (UBAP2 localization in siRNA exp, Fig S2 D; UBAP2L localization during recovery from stress, Fig 2C; and an arsenite time course showing condensate formation as a function of UBAP2L, Fig S4), as well as corrections and additions to the references. Together, these changes strengthen the evidence for the conclusions we draw and allow a more nuanced interpretation of UBAP2L and stress-responsive condensates. Please find our point-by-point response to the reviewers' comments below.

We are excited to share our revised manuscript with you and trust that it is now suitable for publication at JCB. Thank you for your time and consideration. We look forward to hearing from you soon.

Sincerely,
Claire Riggs

Reviewer #1 (Comments to the Authors (Required)):

In this study, Riggs and his/her co-authors characterize the novel roles of UBAP2L, an RNA-binding protein, in PB biogenesis and SG/PB interaction, besides its known role in SG assembly. They also decipher the protein architecture in PB association. The IF data are beautiful and convincing. The findings are novel and interesting, providing new insights to PB biology and SG/PB association, as well as to RNA metabolism in cells under friendly or adverse conditions. While there are also some concerns worthy of discussion.

Thank you for your positive view of the work and constructive feedback. Your comments have helped us greatly improve the manuscript. Below we respond to each of your comments.

1. Fig.1. UBAP2 also involved in SGs. In Fig.1C, it may be required to examine whether UBAP2 is also associated with PB using a UBAP2 antibody.

We appreciate the reviewer's comment. In U2OS-WT cells (Fig 1C), UBAP2 is not recruited to PBs. We also observe that UBAP2 remains in SGs, regardless of UBAP2L depletion via siRNA. We have added this data to Supplemental Figure 2 (Fig S2 E). These cells are from the same experiment as Fig 1E, but we probed for UBAP2 rather than UBAP2L. These data indicate that a change in UBAP2's localization is not responsible for its lack of functional compensation for UBAP2L. We now include this interpretation in the text (lines 205-207).

2. Fig. 1 aims to show that UBAP2L contributes to the formation of SGs and PBs and their interaction. In Fig. 1D, as UBAP2L is deleted, of course no UBAP2L-positive IF signal was observed, and of course no UBAP2L-SG and no coalescence of UBAP2L-SG and PB marker was observed. It might be more appropriate to use another SG marker for detecting SG/PB association following UBAP2L deletion, to address that UBAP2L contributes to PB formation. In this sense, Fig.2 is well-designed, with G3BP being KO and using UBAP2L as a marker.

We agree. This experiment was originally performed in U2OS-WT cells and UBAP2L KO cells with immunostaining for HEDLS, eIF4G, and eIF3b for the very reason described here by the reviewer. However, we repeated the experiment with the inclusion of the GFP-UBAP2L/UBAP2L KO line to show that UBAP2L reconstitution rescues SG and PB formation and interaction. Since the rescue cell line constitutively expresses GFP-UBAP2L, immunostaining may only be performed on two other channels

(due to microscopy limitations). The original experiments with HEDLS, eIF4G, and eIF3b stand and show the same effect of UBAP2L KO as the data shown in Fig. 1A and quantified in Fig. 1B. Fig. 1A and B demonstrate the change in SG and PB number and interaction *resulting from* UBAP2L KO, which can be observed with or without staining for UBAP2L as long as SG and PB markers are included as we have done. Thus, we feel the current figures we show in Fig. 1A and B are appropriate and the data included here for your review do not merit inclusion in the manuscript.

In Fig. 4, when conclude that G3BP is not required for UBAP2L to form PBs, Fig.3 may also should be cited, as this figure at first use G3BP-KO cells.

Figure 4 shows that G3BP is not required for UBAP2L overexpression to form hybrid granules containing SG and PB components, which is different from forming PBs. Since arsenite is also added to the cells in the experiments shown in Fig 4 we cannot conclude that UBAP2L is driving PB formation here, and thus it does not seem appropriate to refer to PB formation in G3BP KO cells (Fig 3).

4. Like SGs, PBs are also dynamic. It is still unknown whether UBAP2L is present in PBs and contributes to SG/PB interaction when stress is removed or delayed. And what is the contribution of UBAP2L under these conditions?

We have performed an additional experiment to assess UBAP2L localization during recovery from stress, which now appears in Fig. 2C. Here we observe that UBAP2L shifts from SGs to PBs over the course of recovery from arsenite treatment. The intensity plot profiles shown below the immunofluorescence images in 2C show UBAP2L aligning with eIF3b (SG marker) peaks up to ~ 30 min after arsenite treatment. After 60 min of recovery, however, UBAP2L aligns with the HEDLS (PB marker) peak on the intensity profile. We interpret recovery to be another situation in which cells experience a mild stress wherein PBs are present but SGs are not. We thank the reviewer for this suggestion as it strengths the conclusions we draw about UBAP2L localization.

Huang et al 2020 concluded that UBAP2L also contributes to SG disassembly. We have added mention of this to the intro (line 54).

5. The significance of the presence of UBAP2L in PBs and SG/PB interaction seems still not clearly characterized, despite of some discussion. Dose it definitely contribute to SG/PB component interchange? Especially RNA for decay in PBs? Component chasing or interchange array in live cells may be a choice to answer this question.

We agree with the reviewer that understanding the significance of UBAP2L in PBs and the SG-PB interaction is an important and exciting direction to pursue, however we feel that illuminating the functions associated with the observed cell biology is beyond the scope of this paper. We do agree that SG-PB functional interaction (beyond UBAP2L) is of utmost importance.

Minor:

1. Table 2, please add the working dilution for WB. Besides, there is a repetition of the information for UBAP2L antibody.

WB dilutions have been added and the antibody information has been corrected.

Reviewer #2 (Comments to the Authors (Required)):

Here the authors suggest that the stress granule component UBAP2L has a role in processing bodies formation. While the manuscript is potentially interesting (more adequate for a report and not a full article), many of the experiments are not quantified, which makes the conclusion an overinterpretation. An additional problem is that the literature is cited incorrectly or not cited at all.

Detailed comments

Two major weaknesses of this manuscript are:

1) the authors only look at one cancer cell line (U2OS)

Based on the editor's feedback, we have chosen not to pursue experiments in additional cell lines. We do, however, reference studies in other cell lines which arrived at the same conclusion regarding UBAP2L's contribution to SG formation.

2) there are no quantifications for any of the major statements that follow figure 1. For example, what is the percentage of colocalization between UBAP2L granules and the other components shown in figure 2 in control and G3BP1/2 KO cells? Same is true for most following figures.

We have added quantification and statistical analysis to several figures, including:

- quantification of siRNA efficiency (WBs) and immunofluorescence data (Fig 1 and Fig S2)
- quantification of protein recruitment to SGs and PBs under different conditions (Fig 2)
- quantification of protein recruitment to UBAP2L foci, supporting their PB identity (Fig 3)
- quantification of PB size and number, HEDLS intensity in SGs, and overlap between PB and SG area in the context of UBAP2L overexpression (Fig 4)
- quantification of formation of hybrid granules in cells expressing different UBAP2L constructs (Fig 5)

There is a lack of citations of literature in model organisms and below I cite a few but not all examples. While it is true that stress induces Processing Bodies formation, PBs in many organisms are always there, independently of any (known) stress (eg *S. cerevisiae*, plants, *Drosophila*, *C. elegans*). Most of the literature cited in the article only refers to cancer cell lines.

Literature in yeast suggests that PBs can be the seeds of SGs, which is very relevant in the context of the data presented here (Hondele et al., Nature, 2019).

Thank you for this suggestion. We have incorporated more references to condensate formation in other models throughout the text. In the intro we have clarified that PBs are often constitutive (~lines 37-39). We have added a section to the results (~lines 476-488) considering condensate interaction and functional significance in *C. elegans* and yeast, including discussion of Hondele et al.

In line 71-72, the authors say that RNAs in PBs lack detectable polyadenylation and cite Hubstenberger et al., 2017. However, in this article it is stated "Thus, poly(A) tails of P-body-enriched mRNAs are not shorter or longer on average, but are more heterogeneous. Altogether, P-body-targeted mRNAs are characterized by variable poly(A) tail lengths and poor translation. »

We have corrected the text. In the results (lines 266-268) we now clarify that poly-adenylated mRNAs have not been previously detected by *in situ* hybridization and thus our results are consistent with these prior findings in PBs.

Line 130-131, page 5: the authors say that it is unclear if the UBA domain of UBAP2L contributes to SG formation and cite Huang, 2020 and Markmiller, 2018. To my knowledge, Marmiller has no data on the domains of UBAP2L and SG formation while both Huang et al., 2020 and Youn et al, 2018 show that the UBA domain is not required for SG formation.

Fig 6I in Markmiller et al. shows cells expressing either full length UBAP2L or UBAP2L missing the UBA domain. The IF data and their analysis show widespread formation of SGs in the *absence* of stress in cells expressing dUBA-UBAP2L.

Text from results: "...a truncated version lacking the N-terminal ubiquitin-associated UBA domain (DUBA_UBAP2L-mCherry) led to widespread formation of aggregates containing the SG proteins G3BP1, FMR1, and ELAVL1 even in the absence of stress (Figure 6I; Figure S5E)."

Given that Markmiller et al.'s findings show SG formation in unstressed cells upon dUBA-UBAP2L expression and other papers (as you mention) show no effect on SG formation, we conclude that the role of the UBA domain in SG formation is unclear.

Other statements have no reference et al, eg line 60-61, line 132-133

We have added the appropriate citations.

Line 216, page 8: The laboratory of Roy Parker has shown that not only poly(A) mRNAs are in SGs

In line 216 (now line 273) we state that “neither poly(A) mRNA nor PABP are detected in these UBAP2L-positive granules... consistent with their identity as PBs”. We do not feel that this statement conflicts with the reviewer’s comment. We are not claiming that the PBs don’t have *any* RNA and thus are not SGs, but rather that poly(A) mRNA is not visible by ISH, when clearly visible in SGs, suggesting that the foci we are investigating are not SGs.

Line 225, page 8: polysome disassembly is also required for SG formation

This is true. We have added the following text to clarify our conclusion from this experiment:

“While SG formation also depends on polysome disassembly, this further establishes that these foci are not pre-stress seeds (Marmor-Kollet et al., 2020) (Fig 3B), or something else, that forms independent of polysome disassembly.”

Line 246: did Huang et al measure the affinity of UBAP2L and Fragile X proteins?

Huang et al performed an IP showing FXR1 and FXR2 coming down with a Flag-UBAP2L pull down. Sanders et al 2020 also showed FXR1 and FXR2 in GFP-UBAP2L pull downs, which they denoted as a “high affinity interaction” (Fig 2D). However, since it is not clear if/how affinity was measured, we have revised the manuscript to read as follows:

“G3BP and Fragile X proteins, which were shown to interact with UBAP2L by immunoprecipitation (Huang et al., 2020; Sanders et al., 2020)”

Minor comments:

Lines 157-158: the authors state: “proximity labelling studies only identified the involvement...”. Proximity labelling studies do not identify involvement, they only identify proteins (or other molecules) proximal to the bait.

We have revised the text as follows: *“Additionally, unlike UBAP2L, UBAP2 was not identified in proximity to key SG proteins prior to stress”*

Lines 192-193: A review is cited to support the statement

Appropriate primary sources have been added. We have kept the review as it nicely summarizes this information and may be helpful to readers.

Line 265: here also the references seem a bit random

Huang et al show G3BP and FXR1 bind to UBAP2L and Sanders et al. show UBAP2L binding to DDX6 in

addition to FXR1 and G3BP. These are the appropriate references.

Reviewer #3 (Comments to the Authors (Required)):

The authors show that UBAP2L and its homolog UBAP2 both localise to SG in oxidative stress. However, KD of UBAP2 does not seem to modulate SG formation, so the authors focused on UBAP2L. KO of UBAP2L abolishes SG formation and surprisingly also reduces PB formation and PB-SG docking; expression of GFP-UBAP2L restores these defects. In cells lacking SG (WT with mild osmotic stress, or G3BP1/2 KO with arsenite) UBAP2L also localises to PB. As determined by IF, the foci formed in these SG-lacking cells contain known PB marker proteins (including proteins known to localise to both PB and SG), but lack SG marker proteins and polyA+ mRNAs. Overexpression of UBAP2L induces formation of granules that contain both PB and SG marker proteins in stress, but mostly PB components without stress. The authors further identify region 205-291, containing two RNA binding domains, as essential for PB-SG docking and merging.

The paper is very well written, and figures clearly labeled, with the introduction giving all background required even for non-experts, and the discussion putting the results nicely into context of the published literature.

Overall, the only major comment I have is that appropriate quantification of protein enrichment would greatly help the reader evaluate the original data, especially since in most cases only few cells are shown.

We appreciate this reviewer's comments on the significance and the quality of the paper itself. We completely agree on the importance of the quantification issue. We have added quantification and statistical analysis to several figures, including:

- quantification of siRNA efficiency (WBs) and immunofluorescence data (Fig 1 and Fig S2)
- quantification of protein recruitment to SGs and PBs under different conditions (Fig 2)
- quantification of protein recruitment to UBAP2L foci, supporting their PB identity (Fig 3)
- quantification of PB size and number, HEDLS intensity in SGs, and overlap between PB and SG area in the context of UBAP2L overexpression (Fig 4)
- quantification of formation of hybrid granules in cells expressing different UBAP2L constructs (Fig 5)

We have also added images in figure 4A showing more cells to see the overall effect of UBAP2L overexpression on the nature and relationship of SGs and PBs to each other.

Figure S1: siUBAP2L seems to be reduced by siUBAP2. Please quantify and, if true, please mention in text or figure legend of the main figure.

Quantification of siRNA efficiency for 3 independent replicates has been added to the supplemental. See fig. S2 A, B. siUBAP2 did not significantly reduce UBAP2L levels.

Figure 1C: the blue eIF3b signal as stress granule marker is hard to see in the merged images, the UBAP2L (red) signal is very strong. It would be good to have the split channel images, at least as a

supplementary figure, to be able to judge the changes in the eIF3b signal. Please add a quantification (as in Figure 1E).

Thank you for these suggestions. siRNA images shown in Fig. 1E, Fig. S2 C, and Fig. S2 D now include an enlarged area shown separately for each channel. Quantification for the siRNA immunofluorescence data has been performed and now appears in Fig. 1F.

Figure 2A: as the authors write at the end of the paragraph, recruitment of UBAP2L to PB seems to be not as efficient in sorbitol; some kind of quantification, e.g. intensity measurements along a line through the MLO' for all three marker proteins would be very helpful.

Thank you for these suggestions. We have revised figure 2 to focus on one example (2A from the original submission) and now show intensity profile plots through SGs and PBs and quantification of protein recruitment to SGs and PBs in the different conditions, allowing for statistical comparison of differences in protein recruitment.

Figure 3A: maybe color code the legend to help the reader (e.g. write "PB components" in green). The 'shared' component YB1 seems not to be recruited, also eIF4E seems to be less enriched than others; therefore I would suggest to find a way to quantify this data.

We have added intensity profile plots through foci in figure 3A and added quantification of each protein's recruitment to the foci of interest (3B). Fig 3B has been color coded, however this proves challenging for Fig 3A since we were not able to be entirely consistent in immunofluorescence color assignments. For example, for PB antibodies raised in rabbit (eg XRN1), we could not use it along with the UBAP2L antibody (also rabbit) so it was necessary to use HEDLS to denote the foci of interest, which we have consistently color-coded green, preventing XRN1 from also appearing in green. However, the addition of 3B clearly illustrates the differences in recruitment for each protein investigated and the proteins grouped by category, which we trust will greatly aid the reader in interpreting this figure. Regarding the low recruitment of YB1, this is correct and shown now by quantification. Though it is categorized as a shared protein, this is because it relocalizes to PBs under certain conditions, rather than always presenting in both SGs and PBs. Thus this variation within the 'SG/PB proteins' is anticipated.

Figure 4A / 5A, 24h time point: While the SG marker proteins (UBAP2L, eIF4E, eIF4G, PABP) seems to form one large granule, the PB marker proteins (HEDLS, DDX6, EDC3 etc) seems to form foci within this granule. Could the authors please detail why they consider this 'uniform distribution' / or 'merged'? I would suggest to rephrase because at least to me, the term 'uniform' implies a fully mixed granule.

Thank you for this comment. We decided to re-phrase to refer to the granules formed by UBAP2L overexpression as 'hybrid granules', encompassing proteins that appear homogenous in their distribution through the granule as well as PB proteins that appear in PB-like foci or puncta. Hybrid suggests a mixture of the two types of granules, which is what we observe, rather than proteins simply re-localizing to a different type of condensate.

Figure 5A/5B: please quantify, and if possible color-code again PB and SG and shared marker proteins to make it easier for the reader.

In revising the manuscript, we decided to de-emphasize characterization of all the proteins in the hybrid granules. Figure 5A and B have been moved to the supplemental and now appear as Figure S5 B. We

have added symbols to the photos in this figure to denote which proteins are SG, PB, or SG/PB-localizing proteins to aid the reader.

In Figure 4, we added quantification showing the change in the number and size of PBs in relation to increasing levels of UBAP2L. We also quantified the intensity of HEDLS inside the boundary defined by SGs, as well as the overlap in PB and SG signal, in cells without UBAP2L (0 hrs dox) and cells overexpressing UBAP2L. This quantification in Fig 4 characterizes the important changes we observe with UBAP2L overexpression. Therefore, Fig S5 serves to demonstrate that several PB and SG components are present and that what was observed in Fig 4 is not unique to the proteins selected.

In attempts to quantify recruitment to the UBAP2L-overexpression granules in Fig S5, we encountered some challenges. First, though we can (and did) measure recruitment of different proteins to the UBAP2L granule (see below), it is difficult to draw conclusions from these data since we do not know what is a meaningful level of recruitment.

To attempt to resolve this, we performed additional experiments on WT-U2OS cells for comparison. However, for the most part we did not observe a difference in PB protein intensity in SGs. We suspect this is because of the high spatial overlap – even in WT cells – between PB and SG signal, due to docking PBs. Docking PBs appear just at the edge or ‘on top’ of SGs, creating coincident signal. Based on these challenges. In Fig 4 we performed this analysis on UBAP2L KO (no dox) and UBAP2L overexpressing cells and observed a dramatic difference. In this case docking impairment in UBAP2L KO cells results in minimal overlapping SG/PB area, so the assay is effective.

April 1, 2024

RE: JCB Manuscript #202307146R

Prof. Paul Anderson
Brigham and Woman's Hospital
75 Francis St
Boston, MA 02115

Dear Prof. Anderson,

Thank you for submitting your revised manuscript entitled "UBAP2L contributes to formation of P-bodies and modulates their transient docking to Stress Granules." The manuscript has been re-assessed by all of the original reviewers, whose comments are appended below. We would be happy to publish your paper in JCB pending final revisions necessary to address the remaining reviewer comments and to meet our formatting guidelines (see details below).

You will see that Reviewer #2 asks to explain several apparent discrepancies in quantifications which we agree must be resolved. Reviewer #2 also requests quantification of the extent of colocalization between UBAP2L granules and PBs, this would be an informative addition and hopefully can be done with existing data. The other reviewer comments ask for changes to the text and figures.

A. MANUSCRIPT ORGANIZATION AND FORMATTING:

Full guidelines are available on our Instructions for Authors page, <https://jcb.rupress.org/submission-guidelines#revised>.
Submission of a paper that does not conform to JCB guidelines will delay the acceptance of your manuscript.

1) Text limits: Character count for Articles is < 40,000, not including spaces. Count includes title page, abstract, introduction, results, discussion, and acknowledgments. Count does not include materials and methods, figure legends, references, tables, or supplemental legends.

2) Figure formatting: Articles may have up to 10 main text figures. Scale bars must be present on all microscopy images, including inset magnifications. Molecular weight or nucleic acid size markers must be included on all gel electrophoresis.
- Please add scale bars to Figures S3A/B & S4A. The magnifications in Figures 1A/E, 2A, 3A/C/D, 4E/F, 5B, S2C/D, S3A/B, & S5B should also have scale bars but if adding these is not feasible then you can alternatively state their sizes in the figure legends.
- Please add MW marker labels to the 'total protein gels' in Figure S2A.

Also, avoid pairing red and green for images and graphs to ensure legibility for color-blind readers. If red and green are paired for images, please ensure that the particular red and green hues used in micrographs are distinctive with any of the colorblind types. If not, please modify colors accordingly or provide separate images of the individual channels.

3) Statistical analysis: Error bars on graphic representations of numerical data must be clearly described in the figure legend. The number of independent data points (n) represented in a graph must be indicated in the legend. Please, indicate whether 'n' refers to technical or biological replicates (i.e. number of analyzed cells, samples or animals, number of independent experiments). If independent experiments with multiple biological replicates have been performed, we recommend using distribution-reproducibility SuperPlots (please see Lord et al., JCB 2020) to better display the distribution of the entire dataset, and report statistics (such as means, error bars, and P values) that address the reproducibility of the findings.

Statistical methods should be explained in full in the materials and methods. For figures presenting pooled data the statistical measure should be defined in the figure legends. Please also be sure to indicate the statistical tests used in each of your experiments (both in the figure legend itself and in a separate methods section) as well as the parameters of the test (for example, if you ran a t-test, please indicate if it was one- or two-sided, etc.). Also, if you used parametric tests, please indicate if the data distribution was tested for normality (and if so, how). If not, you must state something to the effect that "Data distribution was assumed to be normal but this was not formally tested."

4) Abstract and title: Please revise these as requested by the reviewers.

5) Materials and methods: Should be comprehensive and not simply reference a previous publication for details on how an experiment was performed. Please provide full descriptions (at least in brief) in the text for readers who may not have access to

referenced manuscripts. The text should not refer to methods "...as previously described."

6) For all cell lines, vectors, constructs/cDNAs, etc. - all genetic material: please include database / vendor ID (e.g., Addgene, ATCC, etc.) or if unavailable, please briefly describe their basic genetic features, even if described in other published work or gifted to you by other investigators (and provide references where appropriate). Please be sure to provide the sequences for all of your oligos: primers, si/shRNA, RNAi, gRNAs, etc. in the materials and methods. You must also indicate in the methods the source, species, and catalog numbers/vendor identifiers (where appropriate) for all of your antibodies, including secondary. If antibodies are not commercial, please add a reference citation if possible.

7) Microscope image acquisition: The following information must be provided about the acquisition and processing of images:

a. Make and model of microscope

b. Type, magnification, and numerical aperture of the objective lenses

c. Temperature

d. Imaging medium

e. Fluorochromes

f. Camera make and model

g. Acquisition software

h. Any software used for image processing subsequent to data acquisition. Please include details and types of operations involved (e.g., type of deconvolution, 3D reconstitutions, surface or volume rendering, gamma adjustments, etc.).

8) References: There is no limit to the number of references cited in a manuscript. References should be cited parenthetically in the text by author and year of publication. Abbreviate the names of journals according to PubMed.

9) Supplemental materials: Articles may have up to 5 supplemental figures and 10 videos but if you need extra space for the quantifications of the colocalization between UBAP2L granules and PBs then you can add another supplemental figure. Please also note that tables, like figures, should be provided as individual, editable files. A summary of all supplemental material should appear at the end of the Materials and methods section. Please include one brief sentence per item.

10) eTOC summary: A ~40-50 word summary that describes the context and significance of the findings for a general readership should be included on the title page. The statement should be written in the present tense and refer to the work in the third person. It should begin with "First author name(s) et al..." to match our preferred style.

11) Conflict of interest statement: JCB requires inclusion of a statement in the acknowledgements regarding competing financial interests. If no competing financial interests exist, please include the following statement: "The authors declare no competing financial interests." If competing interests are declared, please follow your statement of these competing interests with the following statement: "The authors declare no further competing financial interests."

12) A separate author contribution section is required following the Acknowledgments in all research manuscripts. All authors should be mentioned and designated by their first and middle initials and full surnames. We encourage use of the CRediT nomenclature (<https://casrai.org/credit/>).

13) ORCID IDs: ORCID IDs are unique identifiers allowing researchers to create a record of their various scholarly contributions in a single place. Please note that ORCID IDs are required for all authors. At resubmission of your final files, please be sure to provide your ORCID ID and those of all co-authors.

14) JCB requires authors to submit Source Data used to generate figures containing gels and Western blots with all revised manuscripts. This Source Data consists of fully uncropped and unprocessed images for each gel/blot displayed in the main and supplemental figures. Since your paper includes cropped gel and/or blot images, please be sure to provide one Source Data file for each figure that contains gels and/or blots along with your revised manuscript files. File names for Source Data figures should be alphanumeric without any spaces or special characters (i.e., SourceDataF#, where F# refers to the associated main figure number or SourceDataFS# for those associated with Supplementary figures). The lanes of the gels/blots should be labeled as they are in the associated figure, the place where cropping was applied should be marked (with a box), and molecular weight/size standards should be labeled wherever possible. Source Data files will be directly linked to specific figures in the published article.

15) Journal of Cell Biology now requires a data availability statement for all research article submissions. These statements will be published in the article directly above the Acknowledgments. The statement should address all data underlying the research presented in the manuscript. Please visit the JCB instructions for authors for guidelines and examples of statements at (<https://rupress.org/jcb/pages/editorial-policies#data-availability-statement>).

B. FINAL FILES:

Thank you for your attention to these final processing requirements. Please revise and format the manuscript and upload materials within one month. If you need an extension for whatever reason, please let us know and we can work with you to determine a suitable revision period.

Thank you for this interesting contribution, we look forward to publishing your paper in Journal of Cell Biology.

Sincerely,

Karla Neugebauer, PhD
Monitoring Editor
Journal of Cell Biology

Dan Simon, PhD
Scientific Editor
Journal of Cell Biology

Reviewer #1 (Comments to the Authors (Required)):

In the revised version, Riggs and his/her co-authors thoroughly revised their manuscript. They add new data and thorough discussion, and most of the reviewers' comments are substantially addressed. The manuscript is greatly improved. The reviewer still has some comments described below.

1. For the response to comment 1, there seems to be no Fig.S2E in the revised manuscript. Please check it.
2. For the comment 4, the authors add new data (revised Fig.2C) showing UBAP2L localization during recovery from stress, and they stated that UBAP2L shifts from SGs to PBs over the course of recovery. While the second question is still not addressed, that is, what's the biological significance of this location shift, i.e. component exchange for SG disassembly? What would happen to PB dynamics and PB-SG association following UBAP2L inactivation? These questions are important, although the authors stated that these questions are not the scope of the manuscript. Anyway, the present version is acceptable. The reviewer looks forward to the subsequent interesting findings from the authors in the future to thoroughly investigate these questions.
3. In the revised version, the title is changed to "UBAP2L contributes to formation of P-Bodies and modulates their transient docking to Stress Granules". The first part of the title is convincing. For the second part, the reviewer has following comments. The word "docking" has two layers of meaning: 1) PBs docks to SGs just like a ship docks to a harbor. The harbor is bigger than the ship, analogizing SGs to PBs. 2) SGs are formed first or relatively immobile, and PBs are formed later, but are active and move to SGs for physical and/or functional interactions, just like a ship docking to a harbor for cargo loading and unloading.

These two issues are interesting and important, which would greatly help us to understand both the SG and PB biology. While the present evidences from the manuscript seem answer neither of above comments. In this sense, "UBAP2L contributes to formation of P-Bodies and modulates their association with Stress Granules" might be more appropriate. These comments are for discussion only, and the authors may not necessary make a change.

4. The Fig.6, model figure, it would be better to keep constant the color for each object, i.e. the schematic graph for UBAP2L, for easily tracing the change of proteins under different conditions. More important, there seems to be no data in the manuscript to support the description that when no DDX6, UBAP2L moves to G3BP granules. If this is drew from other literatures?

Reviewer #2 (Comments to the Authors (Required)):

This revised version of the article has much improved. My comments are mostly about suggested text changes but there is one problem with the paper, now that the data have been quantified, which is not a minor issue.

I have a concern with Figure 1: in 1B the authors show that in UBAP2L KO the average number of SG/cell is reduced, consistent with several previously published articles, in the KO or in depleted cells. However, in Figure 1F, the number of SG/cell in siUBAP2L is actually higher than in the control siRNA..... and the average PB per cell is not less either, inconsistent with the data shown in figure 1B. Since the RNA interference works well (Fig. S1B), what is the problem here? Also, the effect of UBAP2L depletion has been published by several laboratories. Either there is a problem with the experiment itself or, most likely from the image the authors show and the comments in the text, the quantifications are a problem. If the quantifications are an issue, other quantifications may be questionable as well.

Indeed, concerning the quantifications, in page 7 the authors write: Measurements of percent recruitment of UBAP2L, HEDLS, and eIF3b to SGs and PBs in response to different treatments show similar levels of UBAP2L recruitment to SGs in U2OS-WT cells and to PBs in G3BP1/2 KO cells.

However, when looking at Figure 2 A, while in SG UBAP2L is increased from less than 100 relative intensity to about 250 relative intensity, in the PB this goes from more than 100 to just about 200 (comparing arsenite treatment, for example). How is it possible that the calculated enrichment in 2B is the same?

Concerning these quantifications, it would have been more interesting to show how many UBAP2L granules colocalize with PB and viceversa. Is it 100% or less? Are there PBs without UBAP2L and viceversa, are there UBAP2L granules without a PB marker?

Minor comments:

As the authors themselves say in line 38-40, "Most human cell types and organisms display constitutive PBs...." suggesting that the cell line they use, U2OS, is actually an exception. Because of this, the conclusions should be more careful and rigorous. For example, in the abstract, the authors should clearly say: " We find that UBAP2L is not solely a SG protein, but also localizes to PBs in certain conditions, contributes toPB biogenesis, to SG/PB interactions, and can nucleate hybrid granules containing SG and PB components in U2OS cancer cells. These findings inform a new model for SG/PB formation in the context of UBAP2L's role in this cancer cell line.

Also, at the beginning of the abstract, the authors say that stress triggers the formation of PBs. Again, in many cells PBs are there independently of stress (and actually, as mentioned above, U2OS seems rather an exception as HeLa, HEK293, glioblastoma cell lines and more do have PBs independently of stress). This should be rephrased.

The way to answer the question if the UBA domain is required for SG formation is to see whether a construct lacking the UBA domain can rescue or not SG formation. This has been done by Huang et al and Youn et al. In both papers, a construct lacking the UBA domain can rescue SG formation to a level that is almost as in control. This suggest that the UBA domain may contribute but is certainly not required for SG formation. The data in Markmiller do not answer this question. They show that overexpression of a construct lacking the UBA domain can result in phase separation of the protein overexpressed (and that this can bind to other SG components). If anything, this shows that the UBA domain is not required for LLPS.

Reviewer #3 (Comments to the Authors (Required)):

The authors have addressed my comments.

Some small corrections I would suggest

Figure 1E: please label the small zoom-in boxes; eIF3b is blue on top of the figure but grey-scale for the 'inserts'.

Figure 2A: have these been acquired at the same laser settings ? Looking at the same protein across the different panels, the he background fluorescence as quantified is quite different.

Figure 3A, list of Dual SG/PB components at top of the figure: there is a space missing between the dot and 4E-T

2nd Revision - Authors' Response to Reviewers: May 29, 2024

Dear Drs. Neugebauer and Simon,

We are pleased to share our revised manuscript with you and look forward to its publication in JCB. We appreciate the concerns raised and have thoroughly addressed them. Thank you for your patience awaiting our revision.

To address inconsistent quantifications from CRISPR KO and siRNA data, we developed and coded a reproducible analysis pipeline in FIJI. This allowed us to consistently re-analyze all data (in case there were inconsistencies in our original manual counting) and identify two independent experiments with odd results, which we were able to determine was a result of high cell density. Overly dense cells reduce the visible area of the cytoplasm in each cell, erroneously altering SG and PB counts. We thus repeated 2 of the siRNA experiments and controlled for cell number upon re-seeding cells for immunofluorescence. Using the same new analysis code we obtained results consistent with the observations from the UBAP2L KO cells – which were also reanalyzed with the new code.

For the second quantification issue of inconsistencies between intensity profile plots and protein recruitment quantification, after repeating and re-analyzing experiments, we have chosen to present the data slightly differently. It is difficult to meaningfully interpret levels of protein enrichment within PBs or SGs, so we have opted to exclude such data. Since the goal was to simply show that UBAP2L is recruited to PBs in certain conditions, we prefer to present these data as the number of PBs or SGs containing UBAP2L, as suggested by you and a reviewer. To determine if UBAP2L is “present” in a given PB or SG, we measured its intensity inside the granule and compared it to its mean intensity in the cytoplasm. Granules with UBAP2L mean intensity at least 3x the mean intensity of the cytoplasm were counted as UBAP2L-positive.

The additional reviewer comments have been addressed below. Manuscript organization and formatting requirements have also been addressed.

We look forward to publishing our work in JCB.

Sincerely,

Claire Riggs

April 1, 2024

RE: JCB Manuscript #202307146R

Prof. Paul Anderson
Brigham and Woman's Hospital
75 Francis St
Boston, MA 02115

Dear Prof. Anderson,

Thank you for submitting your revised manuscript entitled "UBAP2L contributes to formation of P-bodies

and modulates their transient docking to Stress Granules." The manuscript has been re-assessed by all of the original reviewers, whose comments are appended below. We would be happy to publish your paper in JCB pending final revisions necessary to address the remaining reviewer comments and to meet our formatting guidelines (see details below).

You will see that Reviewer #2 asks to explain several apparent discrepancies in quantifications which we agree must be resolved. Reviewer #2 also requests quantification of the extent of colocalization between UBAP2L granules and PBs, this would be an informative addition and hopefully can be done with existing data. The other reviewer comments ask for changes to the text and figures.

A. MANUSCRIPT ORGANIZATION AND FORMATTING:

Full guidelines are available on our Instructions for Authors page, <https://jcb.rupress.org/submission-guidelines#revised>. **Submission of a paper that does not conform to JCB guidelines will delay the acceptance of your manuscript.**

1) Text limits: Character count for Articles is < 40,000, not including spaces. Count includes title page, abstract, introduction, results, discussion, and acknowledgments. Count does not include materials and methods, figure legends, references, tables, or supplemental legends.

2) Figure formatting: Articles may have up to 10 main text figures. Scale bars must be present on all microscopy images, including inset magnifications. Molecular weight or nucleic acid size markers must be included on all gel electrophoresis.

- Please add scale bars to Figures S3A/B & S4A. The magnifications in Figures 1A/E, 2A, 3A/C/D, 4E/F, 5B, S2C/D, S3A/B, & S5B should also have scale bars but if adding these is not feasible then you can alternatively state their sizes in the figure legends.

- Please add MW marker labels to the 'total protein gels' in Figure S2A.

Also, avoid pairing red and green for images and graphs to ensure legibility for color-blind readers. If red and green are paired for images, please ensure that the particular red and green hues used in micrographs are distinctive with any of the colorblind types. If not, please modify colors accordingly or provide separate images of the individual channels.

3) Statistical analysis: Error bars on graphic representations of numerical data must be clearly described in the figure legend. The number of independent data points (n) represented in a graph must be indicated in the legend. Please, indicate whether 'n' refers to technical or biological replicates (i.e. number of analyzed cells, samples or animals, number of independent experiments). If independent experiments with multiple biological replicates have been performed, we recommend using distribution-reproducibility SuperPlots (please see Lord et al., JCB 2020) to better display the distribution of the entire dataset, and report statistics (such as means, error bars, and P values) that address the reproducibility of the findings.

Statistical methods should be explained in full in the materials and methods. For figures presenting pooled data the statistical measure should be defined in the figure legends. Please also be sure to indicate the statistical tests used in each of your experiments (both in the figure legend itself and in a

separate methods section) as well as the parameters of the test (for example, if you ran a t-test, please indicate if it was one- or two-sided, etc.). Also, if you used parametric tests, please indicate if the data distribution was tested for normality (and if so, how). If not, you must state something to the effect that "Data distribution was assumed to be normal but this was not formally tested."

4) Abstract and title: Please revise these as requested by the reviewers.

5) Materials and methods: Should be comprehensive and not simply reference a previous publication for details on how an experiment was performed. Please provide full descriptions (at least in brief) in the text for readers who may not have access to referenced manuscripts. The text should not refer to methods "...as previously described."

6) For all cell lines, vectors, constructs/cDNAs, etc. - all genetic material: please include database / vendor ID (e.g., Addgene, ATCC, etc.) or if unavailable, please briefly describe their basic genetic features, even if described in other published work or gifted to you by other investigators (and provide references where appropriate). Please be sure to provide the sequences for all of your oligos: primers, si/shRNA, RNAi, gRNAs, etc. in the materials and methods. You must also indicate in the methods the source, species, and catalog numbers/vendor identifiers (where appropriate) for all of your antibodies, including secondary. If antibodies are not commercial, please add a reference citation if possible.

7) Microscope image acquisition: The following information must be provided about the acquisition and processing of images:

- a. Make and model of microscope
- b. Type, magnification, and numerical aperture of the objective lenses
- c. Temperature
- d. Imaging medium
- e. Fluorochromes
- f. Camera make and model
- g. Acquisition software
- h. Any software used for image processing subsequent to data acquisition. Please include details and types of operations involved (e.g., type of deconvolution, 3D reconstitutions, surface or volume rendering, gamma adjustments, etc.).

8) References: There is no limit to the number of references cited in a manuscript. References should be cited parenthetically in the text by author and year of publication. Abbreviate the names of journals according to PubMed.

9) Supplemental materials: Articles may have up to 5 supplemental figures and 10 videos but if you need extra space for the quantifications of the colocalization between UBAP2L granules and PBs then you can add another supplemental figure. Please also note that tables, like figures, should be provided as individual, editable files. A summary of all supplemental material should appear at the end of the Materials and methods section. Please include one brief sentence per item.

10) eTOC summary: A ~40-50 word summary that describes the context and significance of the findings for a general readership should be included on the title page. The statement should be written in the present tense and refer to the work in the third person. It should begin with "First author name(s) et al..." to match our preferred style.

11) Conflict of interest statement: JCB requires inclusion of a statement in the acknowledgements regarding competing financial interests. If no competing financial interests exist, please include the following statement: "The authors declare no competing financial interests." If competing interests are declared, please follow your statement of these competing interests with the following statement: "The authors declare no further competing financial interests."

12) A separate author contribution section is required following the Acknowledgments in all research manuscripts. All authors should be mentioned and designated by their first and middle initials and full surnames. We encourage use of the CRediT nomenclature (<https://casrai.org/credit/>).

13) ORCID IDs: ORCID IDs are unique identifiers allowing researchers to create a record of their various scholarly contributions in a single place. Please note that ORCID IDs are required for all authors. At resubmission of your final files, please be sure to provide your ORCID ID and those of all co-authors.

14) JCB requires authors to submit Source Data used to generate figures containing gels and Western blots with all revised manuscripts. This Source Data consists of fully uncropped and unprocessed images for each gel/blot displayed in the main and supplemental figures. Since your paper includes cropped gel and/or blot images, please be sure to provide one Source Data file for each figure that contains gels and/or blots along with your revised manuscript files. File names for Source Data figures should be alphanumeric without any spaces or special characters (i.e., SourceDataF#, where F# refers to the associated main figure number or SourceDataFS# for those associated with Supplementary figures). The lanes of the gels/blots should be labeled as they are in the associated figure, the place where cropping was applied should be marked (with a box), and molecular weight/size standards should be labeled wherever possible. Source Data files will be directly linked to specific figures in the published article.

15) Journal of Cell Biology now requires a data availability statement for all research article submissions. These statements will be published in the article directly above the Acknowledgments. The statement should address all data underlying the research presented in the manuscript. Please visit the JCB instructions for authors for guidelines and examples of statements at (<https://rupress.org/jcb/pages/editorial-policies#data-availability-statement>).

B. FINAL FILES:

-- Cover images: If you have any striking images related to this story, we would be happy to consider

them for inclusion on the journal cover. Submitted images may also be chosen for highlighting on the journal table of contents or JCBhomepage carousel. Images should be uploaded as TIFF or EPS files and must be at least 300 dpi resolution.

****It is JCB policy that if requested, original data images must be made available to the editors. Failure to provide original images upon request will result in unavoidable delays in publication. Please ensure that you have access to all original data images prior to final submission.****

****The license to publish form must be signed before your manuscript can be sent to production. A link to the electronic license to publish form will be sent to the corresponding author only. Please take a moment to check your funder requirements before choosing the appropriate license.****

Thank you for your attention to these final processing requirements. Please revise and format the manuscript and upload materials within one month. If you need an extension for whatever reason, please let us know and we can work with you to determine a suitable revision period.

Thank you for this interesting contribution, we look forward to publishing your paper in Journal of Cell Biology.

Sincerely,

Karla Neugebauer, PhD
Monitoring Editor
Journal of Cell Biology

Dan Simon, PhD
Scientific Editor
Journal of Cell Biology

Reviewer #1 (Comments to the Authors (Required)):

In the revised version, Riggs and his/her co-authors thoroughly revised their manuscript. They add new data and thorough discussion, and most of the reviewers' comments are substantially addressed. The manuscript is greatly improved. The reviewer still has some comments described below.

1. For the response to comment 1, there seems to be no Fig.S2E in the revised manuscript. Please check it.

We apologize for this, the reference given was incorrect and should have been Fig. SD2 (which is now actually Fig. S1D).

2. For the comment 4, the authors add new data (revised Fig.2C) showing UBAP2L localization during recovery from stress, and they stated that UBAP2L shifts from SGs to PBs over the course of recovery. While the second question is still not addressed, that is, what's the biological significance of this location shift, i.e. component exchange for SG disassembly? What would happen to PB dynamics and PB-SG association following UBAP2L inactivation? These questions are important, although the authors stated that these questions are not the scope of the manuscript. Anyway, the present version is acceptable. The reviewer looks forward to the subsequent interesting findings from the authors in the future to thoroughly investigate these questions. We appreciate these comments and agree these are interesting questions to investigate in the future.

3. In the revised version, the title is changed to "UBAP2L contributes to formation of P-Bodies and modulates their transient docking to Stress Granules". The first part of the title is convincing. For the second part, the reviewer has following comments. The word "docking" has two layers of meaning: 1) PBs docks to SGs just like a ship docks to a harbor. The harbor is bigger than the ship, analogizing SGs to PBs. 2) SGs are formed first or relatively immobile, and PBs are formed later, but are active and move to SGs for physical and/or functional interactions, just like a ship docking to a harbor for cargo loading and unloading. These two issues are interesting and important, which would greatly help us to understand both the SG and PB biology. While the present evidences from the manuscript seem answer neither of above comments. In this sense, "UBAP2L contributes to formation of P-Bodies and modulates their association with Stress Granules" might be more appropriate. These comments are for discussion only, and the authors may not necessary make a change. We have changed the title as suggested.

4. The Fig.6, model figure, it would be better to keep constant the color for each object, i.e. the schematic graph for UBAP2L, for easily tracing the change of proteins under different conditions. More important, there seems to be no data in the manuscript to support the description that when no DDX6, UBAP2L moves to G3BP granules. If this is drew from other literatures? We are unsure what the reviewer suggests here. If we understand correctly, they are suggesting keeping all SGs the same color and all PBs the same color. With this model we hope to show the changes in protein localization – thus have opted to keep proteins the same color rather than the objects. The colors are meant to indicate the presence of a given protein, or a mixture of two proteins, in SGs or PBs (depending on the case). If we have misunderstood the recommendation here, we are open to changing the coloring. Regarding DDX6, this was a prediction given that we know DDX6 is required for PB formation. If there are no PBs, UBAP2L cannot localize to PBs and thus is predicted to be in SGs with G3BP. We have performed this experiment and see the expected pattern. We can add this data to the supplemental, if necessary.

Reviewer #2 (Comments to the Authors (Required)):

This revised version of the article has much improved. My comments are mostly about suggested text changes but there is one problem with the paper, now that the data have been quantified, which is not a minor issue.

I have a concern with Figure 1: in 1B the authors show that in UBAP2L KO the average number of SG/cell

is reduced, consistent with several previously published articles, in the KO or in depleted cells. However, in Figure 1F, the number of SG/cell in siUBAP2L is actually higher than in the control siRNA..... and the average PB per cell is not less either, inconsistent with the data shown in figure 1B. Since the RNA interference works well (Fig. S1B), what is the problem here? Also, the effect of UBAP2L depletion has been published by several laboratories. Either there is a problem with the experiment itself or, most likely from the image the authors show and the comments in the text, the quantifications are a problem. If the quantifications are an issue, other quantifications may be questionable as well.

See response letter at beginning of document.

Indeed, concerning the quantifications, in page 7 the authors write: Measurements of percent recruitment of UBAP2L, HEDLS, and eIF3b to SGs and PBs in response to different treatments show similar levels of UBAP2L recruitment to SGs in U2OS-WT cells and to PBs in G3BP1/2 KO cells.

However, when looking at Figure 2 A, while in SG UBAP2L is increased from less than 100 relative intensity to about 250 relative intensity, in the PB this goes from more than 100 to just about 200 (comparing arsenite treatment, for example). How is it possible that the calculated enrichment in 2B is the same?

See response letter at beginning of document.

Concerning these quantifications, it would have been more interesting to show how many UBAP2L granules colocalize with PB and viceversa. Is it 100% or less? Are there PBs without UBAP2L and viceversa, are there UBAP2L granules without a PB marker?

See response letter at beginning of document.

Minor comments:

As the authors themselves say in line 38-40, "Most human cell types and organisms display constitutive PBs...." suggesting that the cell line they use, U2OS, is actually an exception. Because of this, the conclusions should be more careful and rigorous. For example, in the abstract, the authors should clearly say: " We find that UBAP2L is not solely a SG protein, but also localizes to PBs in certain conditions, contributes to PB biogenesis, to SG/PB interactions, and can nucleate hybrid granules containing SG and PB components in U2OS cancer cells. These findings inform a new model for SG/PB formation in the context of UBAP2L's role in this cancer cell line.

Changed.

Also, at the beginning of the abstract, the authors say that stress triggers the formation of PBs. Again, in many cells PBs are there independently of stress (and actually, as mentioned above, U2OS seems rather an exception as HeLa, HEK293, glioblastoma cell lines and more do have PBs independently of stress). This should be rephrased.

Changed.

The way to answer the question if the UBA domain is required for SG formation is to see whether a construct lacking the UBA domain can rescue or not SG formation. This has been done by Huang et al and Youn et al. In both papers, a construct lacking the UBA domain can rescue SG formation to a level that is almost as in control. This suggests that the UBA domain may contribute but is certainly not required for SG formation. The data in Markmiller do not answer this question. They show that overexpression of a construct lacking the UBA domain can result in phase separation of the protein overexpressed (and that this can bind to other SG components). If anything, this shows that the UBA

domain is not required for LLPS.
Changed.

Reviewer #3 (Comments to the Authors (Required)):

The authors have addressed my comments.

Some small corrections I would suggest

Figure 1E: please label the small zoom-in boxes; eIF3b is blue on top of the figure but grey-scale for the 'inserts'. With re-coloring of figures, this is no longer applicable.

Figure 2A: have these been acquired at the same laser settings ? Looking at the same protein across the different panels, the he background fluorescence as quantified is quite different. Figure now shows images acquired with the same parameters.

Figure 3A, list of Dual SG/PB components at top of the figure: there is a space missing between the dot and 4E-T. Corrected.